# A Generic First-Order Radiative Transfer Modelling Approach for the Inversion of Soil and Vegetation Parameters from Scatterometer Observations

**Raphael Quast [1],\* , Clément Albergel [2] , Jean-Christophe Calvet [2] and Wolfgang Wagner [1]**

[1] Department of Geodesy and Geoinformation, TU Wien, Gußhausstraße 27-29, 1040 Vienna, Austria;
Wolfgang.Wagner@geo.tuwien.ac.at

[2] CNRM—Université de Toulouse, Météo-France, CNRS, 31057 Toulouse, France;
clement.albergel@meteo.fr (C.A.); jean-christophe.calvet@meteo.fr (J.-C.C.)

\* Correspondence: raphael.quast@geo.tuwien.ac.at; Tel.: +43-58801-12267

**Abstract:** We present the application of a generic, semi-empirical first-order radiative transfer modelling approach for the retrieval of soil- and vegetation related parameters from coarse-resolution space-borne scatterometer measurements ($\sigma_0$). It is shown that both angular- and temporal variabilities of ASCAT $\sigma_0$ measurements can be sufficiently represented by modelling the scattering characteristics of the soil-surface and the covering vegetation-layer via linear combinations of idealized distribution-functions. The temporal variations are modelled using only two dynamic variables, the vegetation optical depth ($\tau$) and the nadir hemispherical reflectance ($N$) of the chosen soil-bidirectional reflectance distribution function ($BRDF$). The remaining spatial variabilities of the soil- and vegetation composition are accounted for via temporally constant parameters. The model was applied to series of 158 selected test-sites within France. Parameter estimates are obtained by using ASCAT $\sigma_0$ measurements together with auxiliary Leaf Area Index ($LAI$) and soil-moisture ($SM$) datasets provided by the *Interactions between Soil, Biosphere, and Atmosphere* (ISBA) land-surface model within the SURFEX modelling platform for a time-period from 2007–2009. The resulting parametrization was then used used to perform $SM$ and $\tau$ retrievals both with and without the incorporation of auxiliary $LAI$ and $SM$ datasets for a subsequent time-period from 2010 to 2012.

**Keywords:** remote sensing; microwave; radar; Advanced Scatterometer (ASCAT); soil moisture; radiative transfer; vegetation; backscatter model

## 1. Introduction

The usage of microwave backscatter measurements for deducing biophysical characteristics of the land surface is a well known and widely accepted approach [1–3]. To distinguish contributions that originate from the soil-surface from those originating from the vegetation, a description of the scattering-behaviour of the soil- and vegetation composition within the illuminated area is necessary. The presented investigation is focussed on the use of satellite-borne scatterometer measurements. Therefore, both the coarse spatial resolution (in the order of kilometres) and the limited amount of computational complexity (that can be used to perform large-scale simulations over long time-periods) must be taken into account within the development of a backscattering model. Consequently, considering the coarse resolution of scatterometers and the limited number of independent observables, it is evident that the functional representation of the scattering behaviour must inevitably subsume many different aspects of both soil and vegetation into a constricted set of parameters. To arrive at a suitable description of the scattering mechanisms, it is common practise to use well-defined, idealized objects such as randomly oriented dielectric cylinders and discs to model

the vegetation scattering-behaviour [4–6] and random rough surfaces [7,8] together with dielectric mixing models (like [9]) to address bare soil-scattering. While such modelling approaches provide valuable insights when being applied to well-defined experimental conditions, their application to coarse-resolution space-borne scatterometer measurements results in a set of parameters that are difficult to connect to accessible properties of the observed scene. Furthermore, since the calculation of scattering characteristics based on pre-defined geometrical objects requires specification of their geometry, orientation, and location distributions as well as dielectric properties, such models usually incorporate a large number of parameters that allow a wide range of modelling possibilities. Using measurements with a ground-resolution in the order of kilometres, neither the assessment nor the interpretation of such parametrizations can be performed in a well-defined manner without immense efforts to characterize each site individually. Consequently, the initially physically-based parameters of the backscattering model are in the end treated merely as fit-parameters that are used to empirically adjust the scattering pattern such that the model is capable of representing the observed measurements. Therefore, when considering the practical aspects of this problem, the generally rather complex mathematical formulation resulting from the specification of geometrical structures will mostly be an obstacle to the semi-empirical calibration of the model, since after all, the retrieved parameter values are anyway hardly connectible to identifiable properties of the observed scene. In this paper, we present a generic first-order radiative transfer model parametrization framework that can be regarded as a generalization of the water-cloud model [10]. The specification of the scattering behaviour of both soil- and vegetation is achieved by using linear combinations of idealized scattering distribution functions. The resulting model can thus use both angular-and temporal variations of the backscattering measurements in order to deduce parameters related to soil- and vegetation properties. Furthermore, the impact of first-order interaction effects can be described in a consistent, analytic way by using the method presented in [11]. In Section 2.1, the specifications and the used functional description of the scattering distributions are given. The used datasets and model-parametrization are addressed in Sections 2.2 and 3, and finally Section 4 presents results of an application of the model to ASCAT backscatter timeseries from 2007–2012 over a set of 158 test-sites representing the main agricultural and forest regions in France. Due to the fact that an a priori parametrization of the scattering characteristics of a rough soil surface as well as a covering vegetation-layer would require detailed information on each individual site that is not accessible within the scope of the presented investigation, we focus on analysing possibilities for parametrizing the presented model by using an optimization procedure that minimizes the difference between (incidence-angle dependent) ASCAT and modelled $\sigma_0$ datasets based on the following assumptions:

- All parameters except the optical depth of the vegetation layer ($\tau$) and the (nadir) hemispherical reflectance of the soil-surface ($N$) are assumed to remain temporally constant.
- A subset of parameters is chosen to represent the remaining spatial variability of the $\sigma_0$ dataset that is not covered by variations in $\tau$ and $N$. Since the numerical values of those parameters can a priori only be restricted to a physically plausible range, the actual values for each scene are obtained by an optimization-procedure using inputs of auxiliary Leaf Area Index (LAI) (assumed to be $\propto \tau$) and soil-moisture ($SM$) (assumed to be $\propto N$) timeseries provided by the *Interactions between Soil, Biosphere, and Atmosphere* (ISBA) land-surface model within the SURFEX modelling platform [12].
- The remaining (spatially and temporally constant) parameters are set based on empirical adjustments to achieve a reasonable agreement between ASCAT and modelled $\sigma_0$ for the majority of processed points.

The parameter-values obtained within the calibration-period (2007–2009) are then used to perform a retrieval of daily $SM$ and/or 7-daily $\tau$ estimates for a subsequent time-period (2010–2012).

## 2. Materials and Methods

### 2.1. A Generic, Semi-Empirical Radiative Transfer Modelling Approach

In the following, a generalization of the water cloud model that incorporates a description of the incidence-angle behaviour of the vegetation-scattering pattern and an estimate of first-order interaction-effects according to [11] is introduced. The bare-soil contribution is modelled using parametric functions to represent the Bidirectional Reflectance Distribution Function ($BRDF$) of the soil surface. Furthermore, to be able to account for the fact that the observed scene typically contains both regions of dense vegetation-cover as well as effectively bare soil parts, a parameter accounting for the "effective bare-soil fraction" ($f_{bs}$) is included. The resulting formula for the backscattering coefficient (where the connection of the model-parameters to the biophysical variables has not yet been set) is given by [11] (see Figure 1):

$$\sigma_0 = 4\pi \cos(\theta_0)\left[ \underbrace{f_{bs} \cdot \cos(\theta_0)BRDF}_{\text{bare soil contribution }(\sigma_0^s)} + (1 - f_{bs}) \cdot \left( \underbrace{\gamma^2 \cos(\theta_0)BRDF}_{\text{vegetation covered soil contribution }(\gamma^2\sigma_0^s)} + \right.\right. \tag{1}$$

$$\left.\left. \underbrace{\frac{\omega}{2}\left(1 - \gamma^2\right)\hat{p}}_{\text{vegetation contribution }(\sigma_0^v)} + \underbrace{\sigma_0^{int}}_{\text{interaction contribution}} \right)\right]$$

with the 2-way attenuation factor $\gamma$ defined as:

$$\gamma^2 = e^{-\frac{2\,\tau}{\cos(\theta_0)}} \qquad \text{with} \qquad \theta_0 \ldots \text{viewing zenith angle} \tag{2}$$

The scattering behaviour of bare soil is hereby fully described by the specification of its $BRDF$. The effects of the vegetation are described by the Single Scattering Albedo ($\omega$), the Optical Depth ($\tau$) and the Scattering Phase Function ($\hat{p}$) that accounts for the directionality of scattering within the vegetation layer.

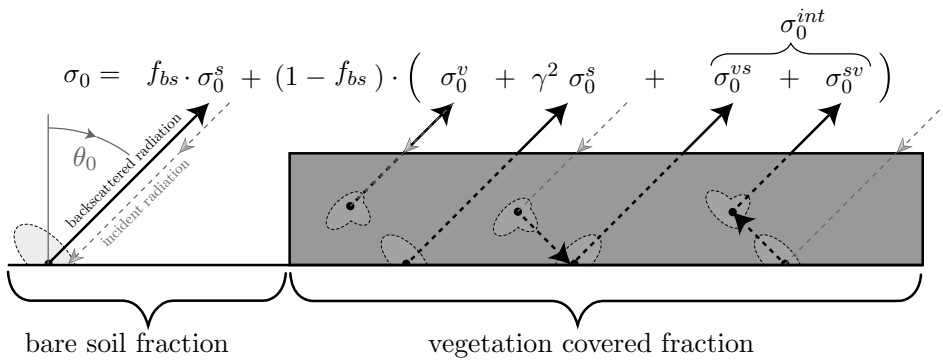

**Figure 1.** Contributions to the backscattered signal.

The additional term $\sigma_{int}^0$ represents first-order (i.e., double-bounce) interaction effects, estimated using approximate representations of $\hat{p}$ and the $BRDF$. Details on the choices for the functional representations of the bistatic scattering characteristics of the vegetation-coverage ($\hat{p}$) and the bare soil surface ($BRDF$) are discussed in the following sections.

### 2.1.1. Parametrization of Scattering Distribution Functions

To approximate the scattering-behaviour of soil- and vegetation, an empirical description of $\hat{p}$ and the *BRDF* based on parametric functions is introduced. While the general framework can be adjusted to mimic a wide range of scattering-characteristics, the final choice for a specific application must be selected with respect to the characteristics of the considered measurements (i.e., frequency, polarization, spatial resolution, . . .). In the presented study, the model is applied to observations from the ASCAT instrument (see Section 2.2.1) where the average illuminated area for each observation is represented by a circle with a radius of approximately 12.5 km. The vegetation scattering-phase-function $\hat{p}$ (as well as $\tau$ and $\omega$) and the *BRDF* of the soil surface thus represent a mixture of physical and structural properties of the vegetation and soil composition within the observed scene. Therefore, the complexity in modelling the scattering characteristics of soil- and vegetation is greatly reduced. The vegetation scattering pattern $\hat{p}$ as well as the *BRDF* of the soil surface are approximated by using linear combinations of generalized Henyey-Greenstein [13] functions $HG(t, \Theta)$.

In its original formulation [13] $HG(t, \Theta)$ has been defined as a single-parametric function (3) that can be used to mimic the scattering pattern of an isotropic ($t = 0$), forward ($0 < t < 1$) or backward ($-1 < t < 0$) scattering target (see Figure 2). The function can be easily expanded in terms of Legendre-polynomials, and is therefore directly applicable for estimating first-order interaction contributions with the method presented in [11]. The functional form of $HG(t, \Theta)$ is given by:

$$HG(t, \Theta) = \frac{1}{4\pi} \frac{1 - t^2}{[1 + t^2 - 2\,t\,\cos(\Theta)]^{3/2}} \tag{3}$$

$$= \frac{1}{4\pi} \sum_{n=0}^{\infty} (2n + 1)\,t^n P_n(\cos(\Theta)) \tag{4}$$

$P_n(x)$ hereby represents the $n$th Legendre polynomial. The scattering-angle $\Theta$ can be expressed in terms of the incident zenith- and azimuth angles $\theta_0, \phi_0$ and the corresponding emergent zenith- and azimuth angles $\theta_s, \phi_s$ via:

$$\cos(\Theta) = -\cos(\theta_0)\cos(\theta_s) - \sin(\theta_0)\sin(\theta_s)\cos(\phi_0 - \phi_s) \tag{5}$$

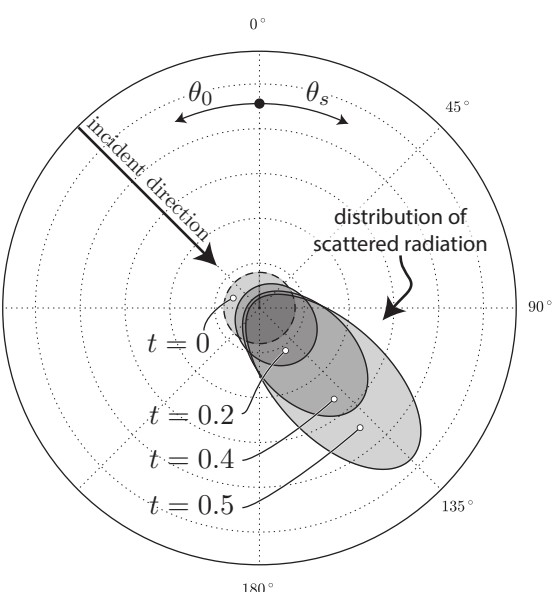

**Figure 2.** Illustration of $HG(t, \Theta)$ [14].

The above definition of the Henyey-Greenstein function can be generalized to represent a broader spectrum of possible scattering distributions by altering the definition of the scattering angle $\Theta$ and introducing an additional parameter $a \in [-1, 1]$ as follows [15]:

$$\cos(\Theta_a) = a \cos(\theta_0) \cos(\theta_s) - \sin(\theta_0) \sin(\theta_s) \cos(\phi_0 - \phi_s) \tag{6}$$

In case $a$ is set to $\pm 1$, the magnitude of the originating scattering pattern remains independent of the difference between the incoming- and scattering angle $(\theta_0 - \theta_s)$. This is illustrated in Figure 3a which shows a linear-combination of 4 generalized Henyey Greenstein functions with the following parametrizations:

$$\hat{p} = \frac{1}{4} \left[ HG_{(A)} + HG_{(B)} + HG_{(C)} + HG_{(D)} \right] \tag{7}$$

where the individual peaks are defined via:

A:　$t = -0.5$　$a = -1$ (backward-peak)　　　　B:　$t = 0.5$　　$a = 1$　(upward-peak)
C:　$t = -0.5$　$a = 1$　(downward-peak)　　　　D:　$t = 0.5$　　$a = -1$ (forward-peak)

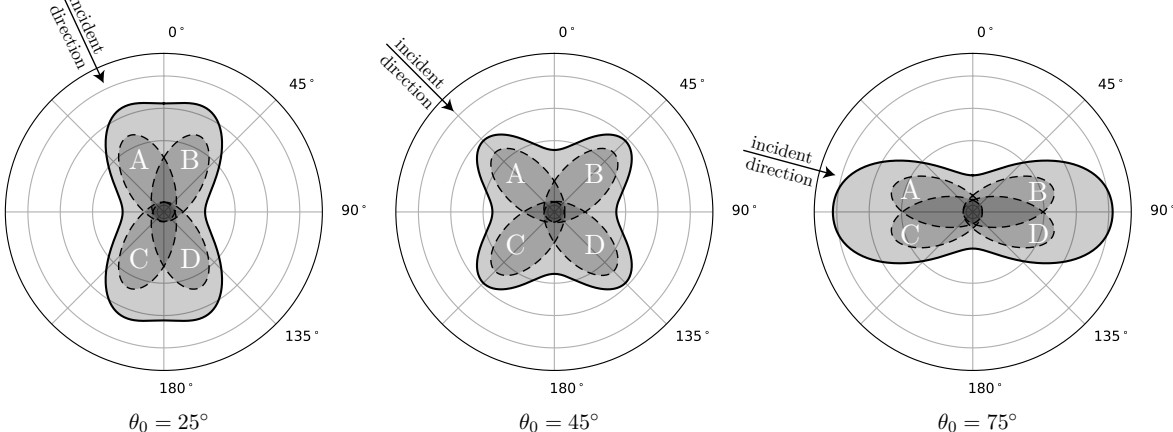

(**a**) Uniform scattering behaviour ($|a| = 1$ for all individual peaks)

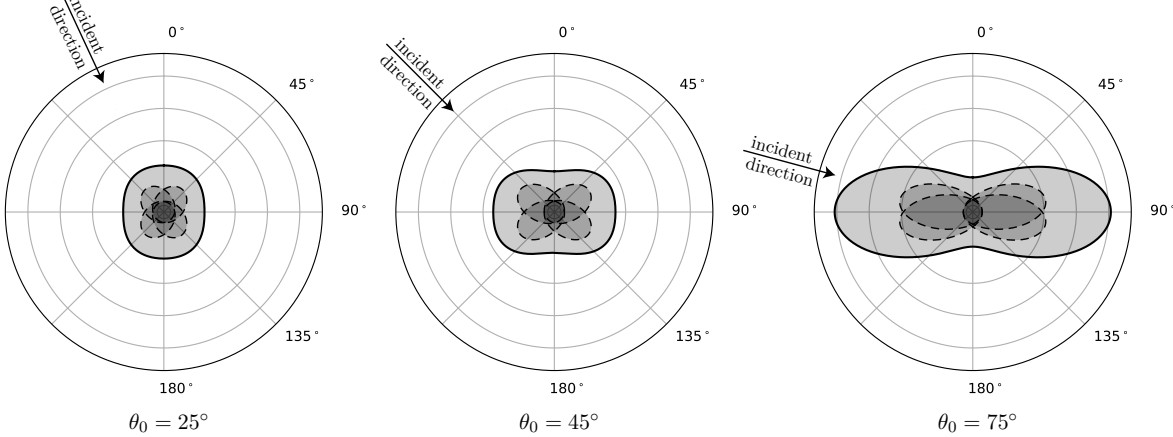

(**b**) Incidence angle dependent magnitude ($|a| = 0.6$ for all individual peaks)

**Figure 3.** Scattering distribution modelled by a linear combination of 4 generalized Henyey Greenstein Functions (see (7)).

If values of $|a| < 1$ are used, the magnitude of the resulting scattering-pattern increases with increasing incidence-angle as illustrated in Figure 3b. While such a parametrization leads to a violation of the normalization condition for volume-scattering phase-functions (see Section 2.1.2), it can be used as a single-parametric way to mimic an incidence-angle dependent normalization for *BRDF* representations as addressed in more detail in Section 2.1.3.

It must be noted that in case exclusively monostatic measurements are used, any contribution from the scattering-patterns away from the monostatic direction will only add to the modelled signal by means of first-order interaction contributions. Therefore, for monostatic retrievals, one does not need to pay too much attention to the full bistatic shape of the pattern.

### 2.1.2. Vegetation Scattering Phase Function $\hat{p}$

As highlighted by de Matthaeis and Lang [16], theoretical calculations suggest that the general shape of the scattering pattern of a dielectric cylinder (at wavelengths in the microwave domain with dimensions that represent average cylindrical vegetation structures) consists of two relative maxima, one in the direction of the incident wave, and one in specular direction. From Figure 3a we see that such a behaviour can for example be approximated by using a combination of peaks similar to cases **C** and **D** as defined by Figure 3a. Using (3), together with ([17], Equation 18.17.6) it can furthermore be shown that $HG(t, \Theta_{(a=\pm1)})$ is already normalized as desired for representing a volume-scattering phase-function, i.e.,:

$$\int_0^{2\pi} \int_0^\pi HG(v, \Theta_{(a=\pm1)}) \sin(\theta_s) d\theta_s d\phi_s = 1 \tag{8}$$

Therefore, any linear combination of generalized Henyey-Greenstein functions (with $a = \pm1$) can directly be used as a well-defined representation for the scattering distribution function $\hat{p}(\theta, \phi)$ of the vegetation-coverage. The specific choice of $HG$-functions that has been used within the presented simulations is addressed in Section 3.3.

### 2.1.3. Parametrization of the Bidirectional Reflectance Distribution Function (*BRDF*)

To define a functional representation that can be used to mimic the scattering behaviour of bare soil, the following general considerations must be considered:

- The overall shape of the *BRDF* is represented by a peak oriented in specular direction whose width is related to the effective roughness and texture of the surface. The term effective is hereby used to indicate that aside of soil-characteristics, also topography as well as other land-cover classes (urban areas, water-bodies . . .) within the observed scene are implicitly subsumed in those parameters.
- In contrast to the scattering-pattern used for representing the vegetation-coverage, the behaviour of Fresnel's reflection coefficients indicate that the amount of scattered radiation in specular direction originating from a (perfectly smooth) surface has a complex (polarization dependent) incidence-angle behaviour. A similar behaviour is expected to be observed when considering the scattering pattern of a rough surface.
- A *BRDF* must be normalized according to [18]:

$$\int_0^{2\pi} \int_0^{\frac{\pi}{2}} BRDF(\theta_0, \phi_0, \theta_s, \phi_s) \cos(\theta_s) \sin(\theta_s) d\theta_s d\phi_s = R(\theta_0, \phi_0) \tag{9}$$

where $R(\theta_0, \phi_0)$ denotes the directional hemispherical reflectance.

A comprehensive incorporation of the aforementioned properties in a functional representation of the *BRDF* requires extensive theoretical calculations [7,8]. However, a parametric description of the *BRDF* that can be used to represent the bare-soil contribution of coarse-resolution monostatic

scatterometer measurements within reasonable accuracy can be obtained by adjusting the previously defined generalization of the Henyey Greenstein function (6) as follows:

$$BRDF(N, t, a) = \frac{N}{R_0(t, a)} HG(t, \tilde{\Theta}_a) \tag{10}$$

The scattering-angle generalization-parameter $a$ is hereby restricted to $0 < |a| < 1$ which allows adjusting the incidence-angle dependency such that the magnitude of the scattering pattern is increasing with increasing incidence-angle (see Figures 4 and 5). To obtain a simple way for specifying the magnitude of the $BRDF$, the normalization-factor $\frac{N}{R_0(t,a)}$ in (10) is defined such that the parameter $N$ represents the hemispherical reflectance at nadir. $R_0(t, a)$ is therefore found by evaluating (9), at $(\theta_0 = 0)$ i.e.,:

$$R_0(t, a) = \int_0^{\frac{\pi}{2}} \int_0^{2\pi} HG(t, \tilde{\Theta}_a) \Big|_{\theta_0=0} \cos(\theta_s) \sin(\theta_s) d\theta_s d\phi_s$$
$$= \frac{(1 - t^2)}{2a^2 t^2} \left[ \frac{(1 + t^2 + at) - \sqrt{(1 + t^2 + 2at)(1 + t^2)}}{\sqrt{1 + t^2 + 2at}} \right] \tag{11}$$

The parameter $t$ thus allows adjusting the scattering pattern from isotropic (very rough surface, $t = 0$) to directional oriented in specular direction (smooth surface, $t = 1$). The parameter $a$ can be used to adjust the incidence-angle behaviour from being uniform ($a = 1$), to increasing with increasing incidence angle ($a < 1$). The parameter $N < 1$ is used to set the nadir-hemispherical reflectance. Additional complexity in the representation of the $BRDF$ (for example the incorporation of an isotropic contribution or peaks in the backscattering direction) can directly be introduced by using linear-combinations similar to (7).

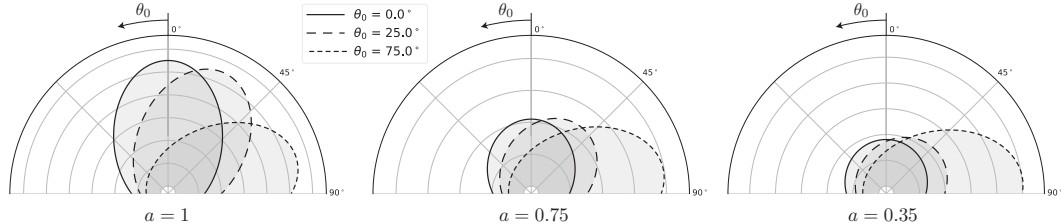

**Figure 4.** Impact of the $a$-parameter on the incidence-angle dependency of the $BRDF$ representation defined in (10). ($t = 0.4$).

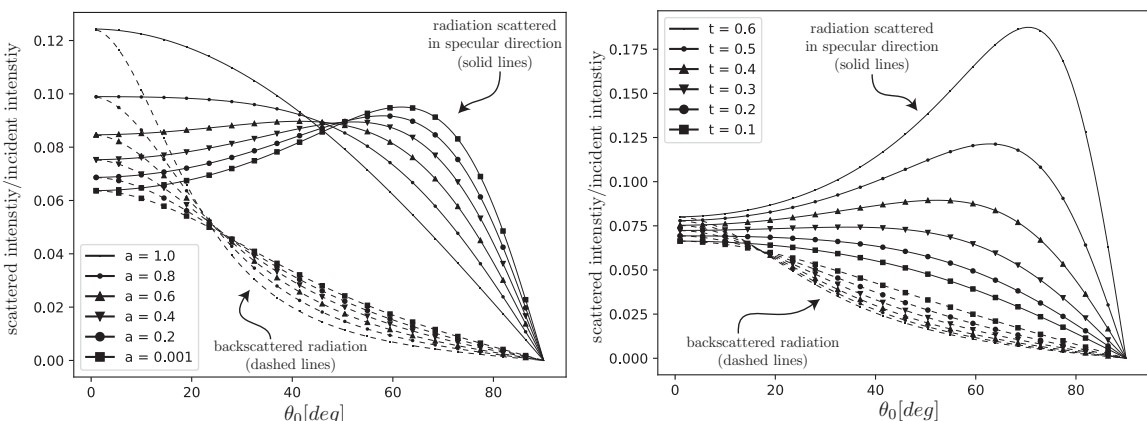

**Figure 5.** Impact of a change in the $a$-parameter (**left**) ($t = 0.4$, $N = 0.2$) and the $t$-parameter (**right**) ($a = 0.4$, $N = 0.2$) on the specular- and backscattered intensity of the $BRDF$ representation defined in (10). $\left( I_s / I_{inc} = cos(\theta) * BRDF(\theta, \phi) \right)$.

## 2.2. Dataset Description

### 2.2.1. ASCAT Backscattering Timeseries

The following investigation is based on backscattering-coefficient ($\sigma_0$) timeseries provided by the Advanced Scatterometer (ASCAT) instrument operated on board the Metop (A) satellite. The measurement-geometry is illustrated in Figure 6, and the main characteristics of the used dataset are summarized below. More detailed information on the measurement principle and the technical description can be found in [19])

- Frequency: C-band (5.225 GHz)
- Polarization: vertical (transmission and reception)
- Swath grid sampling resolution: 12.5 km
- Revisit time: ~1–2 days
- Antenna look angles: ~25° to ~65°

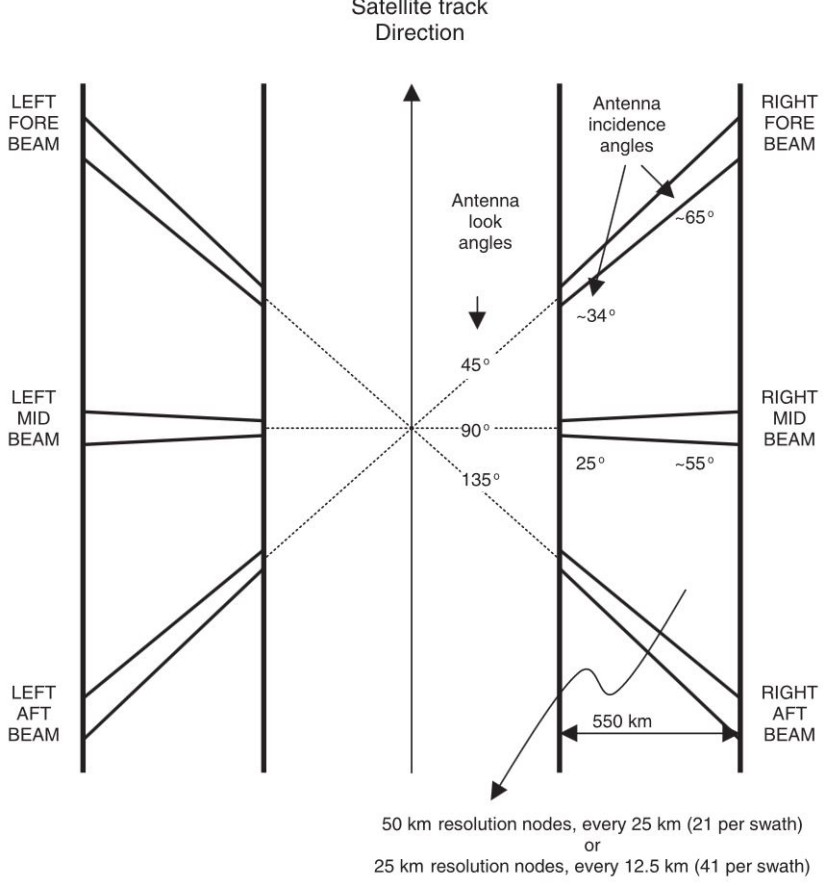

**Figure 6.** ASCAT measurement geometry [19].

The model is applied to timeseries that include measurements from all beams (i.e., FORE, MID and AFT as indicated in Figure 6) as well as ascending and descending orbits in order to maximize the available incidence-angle range as well as the temporal resolution. Furthermore, all days where soil freezing effects are expected have been excluded. The masking is hereby performed based on solid-water content estimates from the top soil layer (~1 cm) as provided by the SURFEX modelling platform [12] alongside the auxiliary *SM*- and LAI datasets.

2.2.2. Auxiliary SM and LAI Datasets

In the present study, the $CO_2$-responsive version of the Interaction between Soil Biosphere and Atmosphere (ISBA) [20,21] Land Surface Model (LSM) forced by the SAFRAN atmospheric analysis is used to compute a soil moisture (surface and root-zone), soil temperature and Leaf Area Index (LAI) database over continental France from 2007 to 2012. SAFRAN (Systeme d'analyse fournissant des renseignements atmospheriques a la neige) [22] is a mesoscale atmospheric analysis system for surface variables. It was initially developed in order to provide an analysis of the atmospheric forcing in mountainous areas for snow depth and avalanche forecasting. The SAFRAN analysis provides the main atmospheric forcing parameters (precipitation, air temperature, air humidity, wind speed, incident radiation) using information from more than 1000 meteorological stations and more than 3500 daily rain gauges throughout France. An optimal interpolation method is used to assign values for each analysed variable. It was shown that a good correlation between the SAFRAN database and in situ observations exists [23].

The $CO_2$-responsive version of the ISBA LSM [24,25] is included in the open-access SURFEX modeling platform of Meteo-France [12]. ISBA simulates the diurnal cycle of water and carbon fluxes, plant growth and key vegetation variables, such as LAI and above ground biomass. Also, in the version used in this study, the hydrology is based on the equations of the force-restore approach [20,21]. The soil layer and soil moisture dynamics are modelled within a 3-soil-layer model [26] with the soil and vegetation parameters being derived from a global database of soils and ecosystems [27] for 12 generic land surface patches which include nine plant functional types (needle leaf trees, evergreen broadleaf trees, deciduous broadleaf trees, C3 crops, C4 crops, C4 irrigated crops, herbaceous, tropical herbaceous, and wetlands), bare soil, rocks, and permanent snow and ice surfaces. Detailed model descriptions can be found in [12]. For the purpose of the land surface simulations, the ISBA parameters, provided by ECOCLIMAP-II [27] at a resolution of 1 km, were aggregated per plant functional type to the model resolution of 8 km. The ISBA model simulation was performed at this resolution using the 'NIT' biomass option. A seven-year simulation was made over France and a subset of 158 points representing the main French agricultural and forest regions has been extracted (as in [28]).

ISBA simulations driven by SAFRAN were compared to in situ observations of SSM in south-western France by Albergel et al. [29] and Draper et al. [30], using 12 stations of the SMOSMANIA [31] observation network. The obtained Spearman temporal correlation coefficient value ranges from 0.6 to 0.8 and the median value is 0.7.

## 3. Choice of Parametrization

In the presented experiments (Section 4), both the choice of the functional representations of $\hat{p}$ and the $BRDF$ as well as the estimation of the associated parameters are based solely on measurements by the ASCAT instrument and simulations of $SM$ and LAI provided by the SURFEX modelling platform. The hereby introduced parametrization is therefore specific to this instrument (ASCAT, see Section 2.2.1) and auxiliary-data model (ISBA forced by SAFRAN, see Section 2.2.2). The overall approach however is highly flexible and can be adjusted to various experiment-configurations. The following section illustrates the considerations that led to the selected model parametrization.

The ASCAT-$\sigma_0$ dataset consists of backscattering measurements with incidence-angles ranging from 25 to 65 degrees, where 3–6 individual measurements at different incidence-angles are available at each day. In an ideal framework, one would set-up the model-parameter estimation as a fully automatic scheme that is capable of retrieving both daily soil-moisture estimates as well as temporally varying estimates for $\tau, \omega$ and the remaining scattering-function parameters. However, considering the available $\sigma_0$ dataset, it is evident that this would result in an extremely under-determined retrieval procedure, where numerous possible parametrizations exist that would all be capable of representing the given daily backscattering measurements within reasonable accuracy. Therefore, the complexity of the parametrization was reduced by representing the temporal dynamics of the dataset exclusively via the optical depth ($\tau$) and the nadir hemispherical reflectance of the $BRDF$ ($N$). All other parameters

that are intended to represent geometrical and structural properties of the observed scene (pixel) are assumed to remain constant over the considered time-period. Furthermore, since the effects of some parameters can be very similar for a monostatic measurement geometry, the fit-procedure is restricted to obtain spatially varying estimates only for the *effective bare-soil fraction* ($f_{bs}$), the asymmetry-factor of the *BRDF* (*t*), the *single-scattering albedo* ($\omega$) and the scaling-factor ($s_2$) that relates soil-moisture (*SM*) and *N* (as defined in (13)). All remaining parameters are kept both spatially and temporally constant, and the numerical values are set based on empirical attempts to adjust forward-simulation and inversion-performances over the chosen test-sites. The resulting dependency of modelled $\sigma_0(\theta_0)$ on the dynamic parameters is illustrated in Figure 7. To show the flexibility of the modelling approach, results for 4 different experiments based on the selected parametrization are shown in Section 4:

(1) forward-simulation of ASCAT $\sigma_0$ timeseries using auxiliary *LAI* and *SM* datasets
(2) *SM* inversion using auxiliary *LAI* datasets
(3) $\tau$ inversion using auxiliary *SM* datasets
(4) simultaneous $\tau$ and *SM* inversion

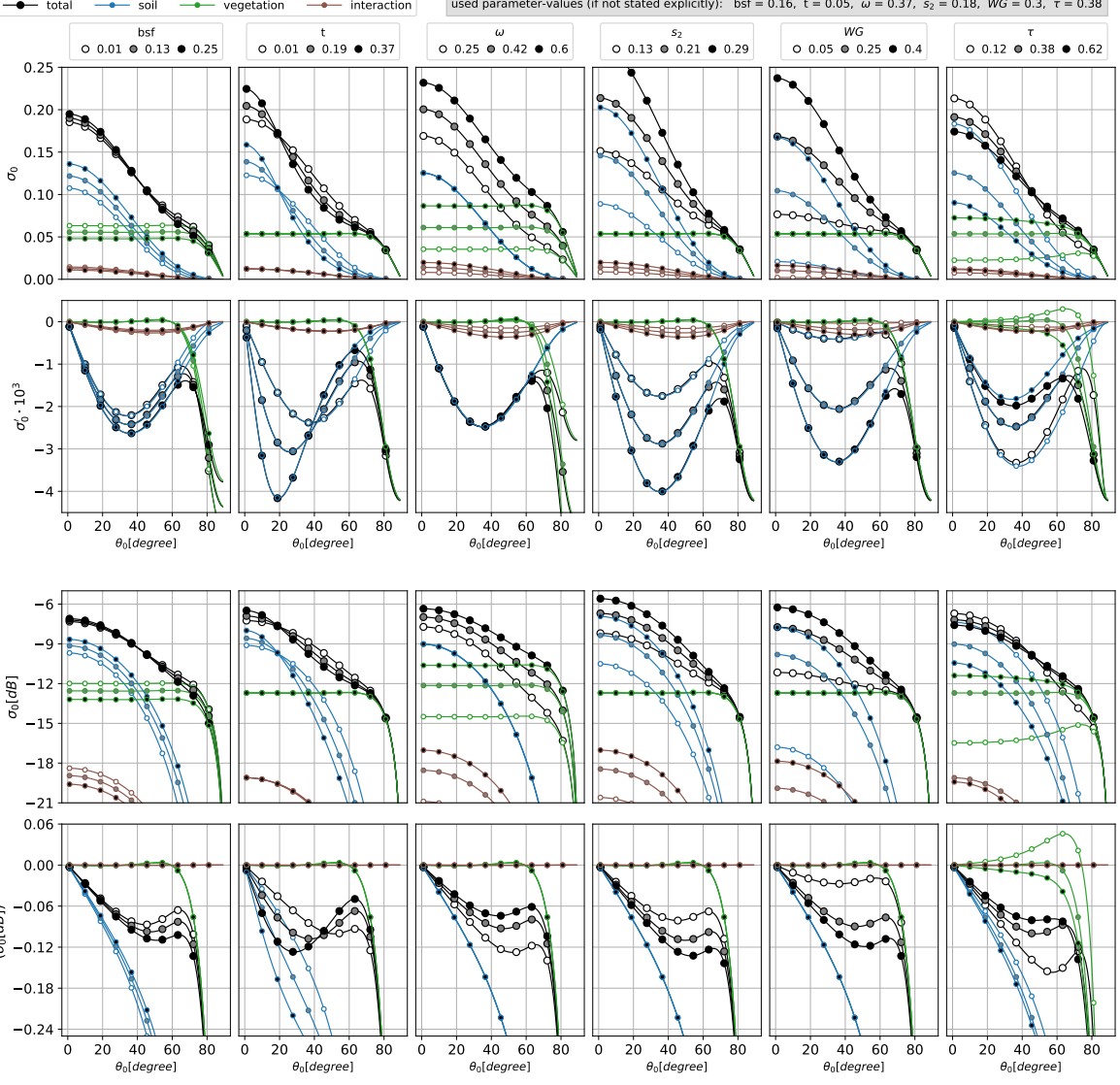

**Figure 7.** Illustration of the behaviour of modelled $\sigma_0$ as well as the resulting soil-, vegetation- and interaction-contributions and the corresponding slopes $\sigma_0' = \frac{d\sigma_0}{d\theta}$ with respect to changes in the model-parameters in linear- (**top**) and dB (**bottom**) scale.

### 3.1. Connection to Biophysical Variables

To connect the results of the above introduced modelling approach to indices that are related to soil- and vegetation properties, a relation between the parameters of the model and the desired biophysical variables is set. Within the following simulation, we investigated the performance of the modelling approach to simulate backscattering coefficient measurements from the ASCAT instrument by using simple empirical relations to connect the optical depth $\tau$ to the leaf-area index $LAI$ and the nadir-hemispherical reflectance $N$ to the soil-moisture content $SM$:

$$\tau = v_2 * LAI \tag{12}$$
$$N = s_2 * SM \tag{13}$$

The numerical values for $s_2$ and $v_2$ are set as follows:

- The range of the parameter $s_2$ is constrained by ensuring that the resulting range of $N$ remains physically plausible. Reported estimates of vertically polarized directional emissivity $\epsilon(\theta)$ at similar frequencies [32–34] (which can be related to the directional hemispherical reflectivity via Kirchhoffs law [35] ($R(\theta) = 1 - \epsilon(\theta)$)) suggest a range of $0 \lesssim N \lesssim 0.1$. Based on the reference-dataset, the range for $SM$ is set to $SM \lesssim 0.45$, and therefore the boundaries for $s_2$ within the fit-procedure were set to $0.1 \leq s_2 \leq 0.3$.
- The connection between $\tau$ and $LAI$ was set to $v_2 = 0.125$, such that an $LAI$ of 8 represents an optical depth of $\tau = 1$ which roughly follows reported ranges of vegetation optical depth estimates at L-, C-, or X-band [36–38].

### 3.2. Numerical Value of the Single Scattering Albedo

Within the literature, many different estimates on the numerical value of the single scattering albedo $\omega$ can be found. Retrieved values (almost entirely based on passive observations) are generally found in the range $0.05 < \omega < 0.15$ [39]. Theoretical models that estimate $\omega$ from assumptions on geometrical and dielectrical properties of the vegetation-constituents usually obtain much higher values. For example, Ferrazzoli et al. [40] obtains values between $\sim 0.4$ to 0.6 for the *"branch-layer of an old forest at L-band ($\theta_0 = 35°$)"*, Xie et al. [41] reports $\omega$-values ranging from 0.3 to as high as 0.8 for *"soy- and cotton fields with varying leave- and stem parameters at C-, X- and Ku-band"* and Liao et al. [42] investigates the dependence of $\omega$ on the vegetation-water content of corn-fields and finds values between 0.4 and 0.6 for V-polarized radiation at 1.26 GHz.

The reason for this discrepancy is that most studies aiming to retrieve soil-and vegetation parameters from passive microwave observations imply so-called "effective single-scattering albedo ($\omega_p$) parametrizations that are defined to implicitly include multiple scattering effects [43]. The numerical value of this effective albedo $\omega_p$ is generally found to range between 0. and 0.15 [40,43,44]. It is important to notice that such effective single-scattering albedo values are strongly dependent on the retrieval-model used for obtaining the values, and connections between the $\omega_p$-values retrieved from passive observations and the actual $\omega$-values are not straightforward as can be seen for example from the works of Kurum et al. [43,45–47].

Within the active remote sensing community, the vegetation coverage is generally modelled using the zero-order "water-cloud model" parametrization as introduced by Attema and Ulaby [10], where the vegetation-contribution ($\sigma_0^v$ in Equation (1)) is parametrized similar to [48]:

$$\sigma_0^v(\theta_0) = A \cdot V_1 \cdot \cos(\theta_0) \left(1 - \gamma^2\right) \qquad \text{with} \qquad \gamma = e^{-\frac{\tau}{\cos(\theta_0)}} \tag{14}$$

Consequently, $\omega$ would hereby be related to the model-parameters $A$ and $V_1$ via:

$$\omega \, \hat{p}_{\text{back}} = A \, V_1 \tag{15}$$

where $\hat{p}_{\text{back}}$ denotes the value of $\hat{p}$ in the backscattering direction (i.e.,: $\theta_s = \theta_0$ and $\phi_s = \pi + \phi_0$).

Thus, stating the actual value of $\omega$ is circumvented by considering only the backscattered part of the scattering pattern and neglecting any multiple-scattering contributions (whose evaluation would require separate estimates of $\omega$ and $\hat{p}$).

The model presented within this paper explicitly includes a parametrization of $\hat{p}$ as well as an estimate of first-order interaction effects. Consequently, the value of $\omega$ is expected to be within the range of theoretical estimates, i.e., $0 < \omega < 0.8$. Since the constitution of the vegetation-coverage is highly dependent on the selected test-sites, the actual value of $\omega$ is determined by the fit-procedure for each site individually.

It must be noted that the choice for $\hat{p}$ and its resulting magnitude in the backscattering direction $\hat{p}(\theta_0 \rightarrow \theta_0)$ determines to a great extent the range of numerical values of the derived $\omega$ estimates. The spatial dynamics of the estimates on the other hand represent actual changes between the $\sigma_0$ datasets that are interpreted as changes in $\omega$ within the retrieval procedure. The reason the parametrization of $\hat{p}$ has been fixed while $\omega$ is retrieved by the fit-procedure stems from the fact that differences in the modelled (monostatic) $\sigma_0$ datasets triggered by a change in $\hat{p}(\theta_0 \rightarrow \theta_0)$ compared to a change in $\omega$ are only evident within the interaction-contributions. The share of those contributions in the total signal however is generally too low to be used as an indicator for the retrieval-procedure to distinguish between the two effects, and consequently when using exclusively monostatic measurements, either $\hat{p}$ or $\omega$ must be set empirically or based on auxiliary information to avoid ambiguous retrievals.

*3.3. Choice of a Vegetation Scattering Phase Function*

The vegetation-scattering phase-function $\hat{p}$ is parametrized using a linear-combination of three *HG*-functions, one representing a forward-scattering contribution ($t = 0.4$), one representing a bounce-off contribution in specular direction ($t = -0.4$) and one representing an isotropic contribution ($t = 0$). The ratio between forward, specular and isotropic contributions was set such that 50% of the radiation is scattered isotropically, and the remaining 50% are split equally between the forward- and specular peak. The number of expansion-coefficients in (3) (used for estimating the interaction-contribution) was set to $n = 8$ for the directional *HG* functions. The resulting shape is illustrated in Figure 8.

$$\hat{p} = \underbrace{0.5 \cdot HG(t=0)}_{\text{isotropic contribution}} + \underbrace{0.25 \cdot HG(t=0.4, a=1)}_{\text{forward-scattering contribution}} + \underbrace{0.25 \cdot HG(t=-0.4, a=-1)}_{\text{specular ('bounce-off') contribution}}$$

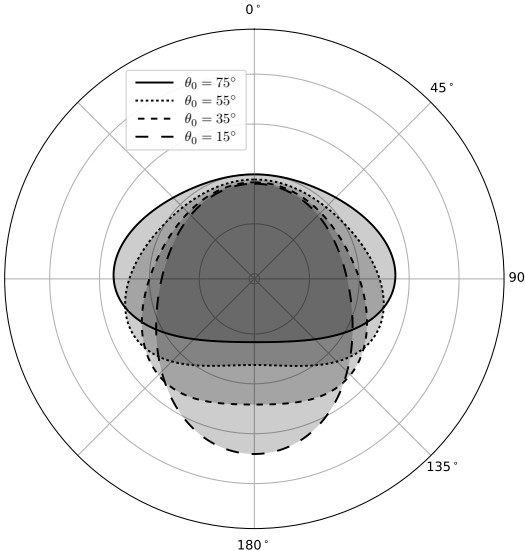

**Figure 8.** Polar-plot of the resulting shape of $\hat{p}$.

### 3.4. Choice of a Surface BRDF

The *BRDF* is directly parametrized using (10). Similarly to the ambiguity between $\hat{p}(\theta_0 \to \theta_0)$ and $\omega$, we find that the impact of the *a*-parameter on the backscattered radiation is not clearly distinguishable from the impact of the *t*-parameter (within the available incidence-angles of the ASCAT-instrument ranging from 25 to 65 degree, see Figure 5). Therefore, we chose to set $a = 0.6$ for the entire set of processed points, and the value of the *t*-parameter is then determined by the fit-procedure for each site individually. The considered numerical range was chosen to be $0 < t < 0.6$, since for $t > 0.6$ soil-scattering would occur mostly in specular direction, while soil-surfaces are generally represented as rough surfaces for observations within the microwave domain. Based on this restriction, the number of expansion-coefficients in (3) (used for computing the estimate of the interaction-term) was set to $n = 10$. The resulting shape of the *BRDF* and the angular dependency of the hemispherical reflectance $R(\theta_0)$ are shown in Figure 9a,b.

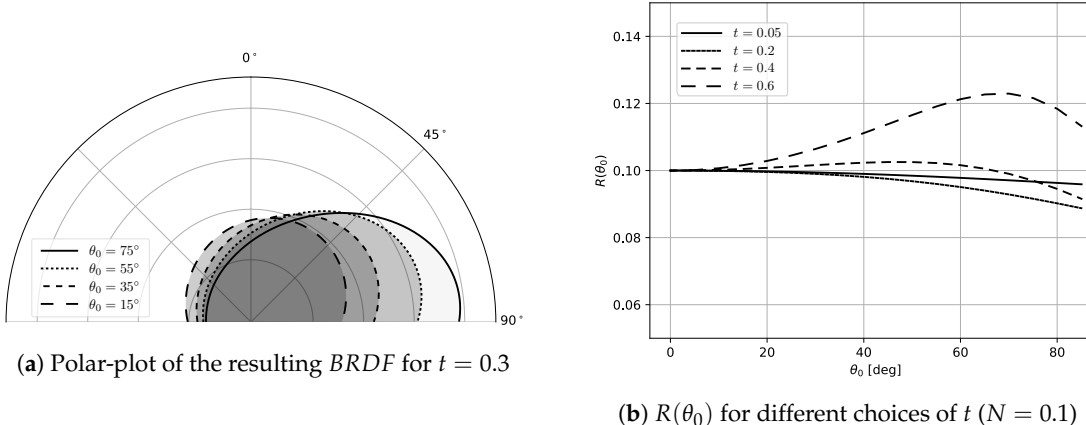

(**a**) Polar-plot of the resulting *BRDF* for $t = 0.3$　　　(**b**) $R(\theta_0)$ for different choices of $t$ ($N = 0.1$)

**Figure 9.** Illustration of the used *BRDF* and its associated hemispherical reflectance $R(\theta_0)$.

### 3.5. Effective Bare Soil Fraction

The $f_{bs}$ parameter serves as a parameter that can be used to enhance the impact of the soil contribution in the total backscattered radiation. Due to the fact that it simultaneously reduces the impact of the volume-contribution, the determination of its numerical value based on an automated least-squares fitting procedure poses some problems:

- In case $f_{bs}$ is too large, a change in the vegetation-parameters will have a rather small impact on the calculated $\sigma_0$, and consequently the obtained vegetation-parameter-estimates from the fit-procedure are very likely to show ambiguity
- If however a sufficient amount of vegetation-contribution is necessary in order to adequately represent the incidence-angle behaviour of the provided $\sigma_0$ dataset, an increase in $f_{bs}$ can be compensated by simultaneously increasing $\omega$. Under certain circumstances this behaviour can lead to a chain-reaction within the fit-procedure that will drive both $f_{bs}$ and $\omega$ to its maximally allowed value.

To avoid the aforementioned problems, the boundaries of the $f_{bs}$-parameter have been set to $0 \leq f_{bs} \leq 0.25$. This choice also reflects the fact that we do not expect areas with rather large fractions of bare-soil within the considered sites.

### 3.6. Fit-Procedure

The model is fitted to ASCAT backscattering observations using a "*Trust Region Reflective*" non-linear least-squares fitting procedure (`scipy.optimize.least_squares`) as implemented within the scientific Python-library *scipy* [49,50]. To reduce computational complexity, an analytic estimation of the Jacobian

at each iteration-step is provided by using a zero-order approximation of the model (i.e., neglecting the interaction contributions). Evaluation of the interaction-term [14] within the calculation of the residuals is performed using the symbolic computation-libraries *sympy* and *symengine* [51] in combination with *numpy* [52]. To increase convergence, a scaling of the individual parameters within the fit-procedure based on *inverse norms of the columns of the Jacobian matrix* as described in [53] has been used.

The used routines are available as part of the (open access) python module RT1 that can be accessed via https://github.com/TUW-GEO/rt1.

### 3.7. Incorporation of Auxiliary Datasets

Estimates for auxiliary LAI- and $SM$-datasets are provided by the SURFEX modelling platform with respect to underlying land-cover classifications within an area of $8 \times 8$ km surrounding the center of the ASCAT grid point location. Even though the SURFEX platform distinguishes more different landcover classes, the presented study considered only estimates of the dominant vegetation classes, i.e.,: **Broadleaf Forests** (BF), **Coniferous Forests** (CF), **Straw Cereals** (SC) and **Grasslands** (GR). In order to gain estimates of LAI and $SM$ that are representative for the average observed scene with respect to ASCAT observations, the provided timeseries are aggregated according to the associated fractional coverages $cov_i$ as follows:

$$LAI = \sum_i \left( \frac{cov_i}{\sum_i cov_i} * LAI_i \right) \qquad SM = \sum_i \left( \frac{cov_i}{\sum_i cov_i} * SM_i \right) \qquad \text{with} \quad i \in [\text{BF, CF, SC, GR}] \qquad (16)$$

The resulting LAI and $SM$ timeseries are then used to force the temporal variability of $\tau$ and $N$ via (12) and (13) as indicated below:

| | Forward Simulation (2007–2009) | $SM$ Inversion (2010–2012) | $\tau$ Inversion (2010–2012) | $SM$ and $\tau$ Inversion (2010–2012) |
|---|---|---|---|---|
| $SM$ input used | YES | NO | YES | NO |
| LAI input used | YES | YES | NO | NO |

## 4. Results and Discussion

In the following section, results for an application of the proposed parametrization to 158 test-sites within France (indicated in Figure 10) are presented. Figure 11 shows density-plots of $\sigma_0^{modelled}$ vs. $\sigma_0^{ASCAT}$ for the forward-simulation (2007–2009) and parameter-inversions (2010–2012). It can be seen that a good overall correlation can be achieved for both forward- and inversion procedures, indicating that the chosen model-parametrization can successfully mimic the dynamics of the ASCAT $\sigma_0$ timeseries. The successful calibration of the model shows that the two completely independent data-sources (i.e., SURFEX simulations and ASCAT $\sigma_0$ measurements) are consistent. The increase in correlation between calibration and inversion results stems from the fact that the least-squares fitting procedure improves the temporal dynamics by adjusting daily $SM$- and/or $\tau$ values. For the forward-modelling experiment, temporal dynamics are predetermined by auxiliary $LAI$ and $SM$ datasets. The fact that no significant increase in correlation can be obtained in the $\tau$-retrieval indicates that the modelled $\sigma_0$ dataset is less sensitive to changes in $\tau$ compared to changes in $SM$. As a consequence, inconsistencies in the auxiliary $SM$ dataset have a greater impact on the retrieved $\tau$ estimate than vice-versa.

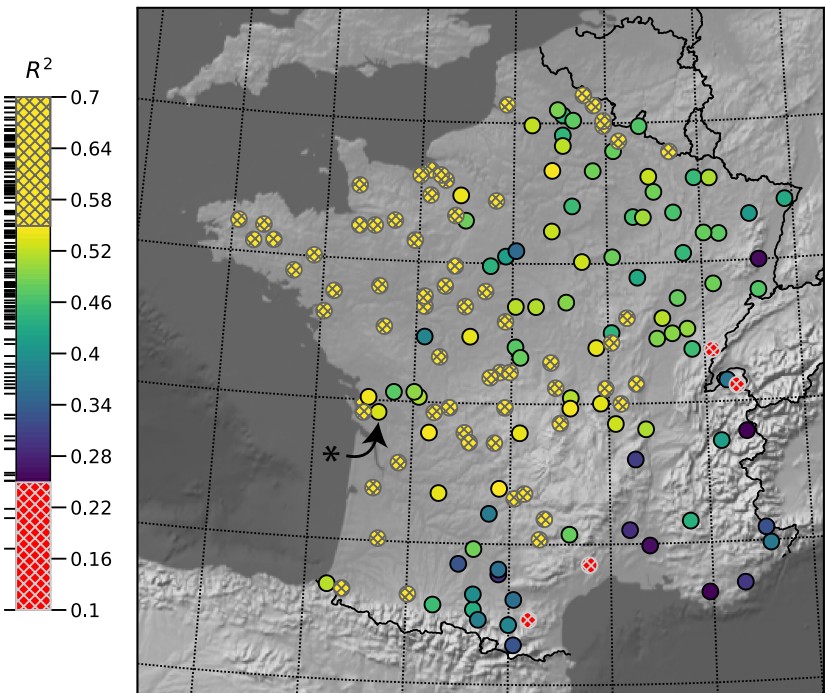

**Figure 10.** $R^2$ of retrieved vs. SURFEX *SM* estimates for retrieval with auxiliary LAI ($\propto \tau$) input. To increase contrast, the color-range is reduced to ($R^2 \in [0.25, 0.55]$). The arrow indicates the site selected in Figures 15 and 16.

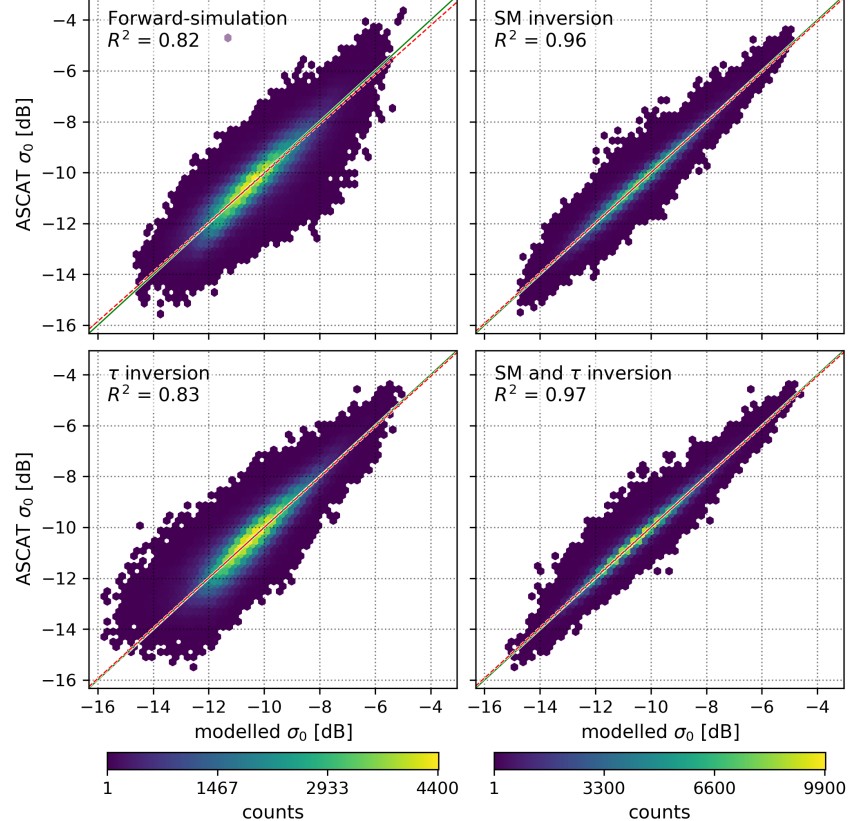

**Figure 11.** Scatterplots of modelled vs. measured $\sigma_0$ in dB for all four experiments. Each plot has been generated using the results from all all processed sites.

Histograms of the resulting parameter-values from the calibration-procedure are depicted in Figure 12. An analysis of the dependencies to fractional vegetation-coverages as well as mean- and standard-deviations of auxiliary *SM*- and LAI timeseries indicated no significant correlations. However, this statement must be treated with care, since the amount of data within this study is rather limited (158 sites), and hardly any of the sites is covered more than 50% by a single vegetation-type. Thus, for a significant conclusion, a larger sample-size and a more rigorous selection of sites would be necessary.

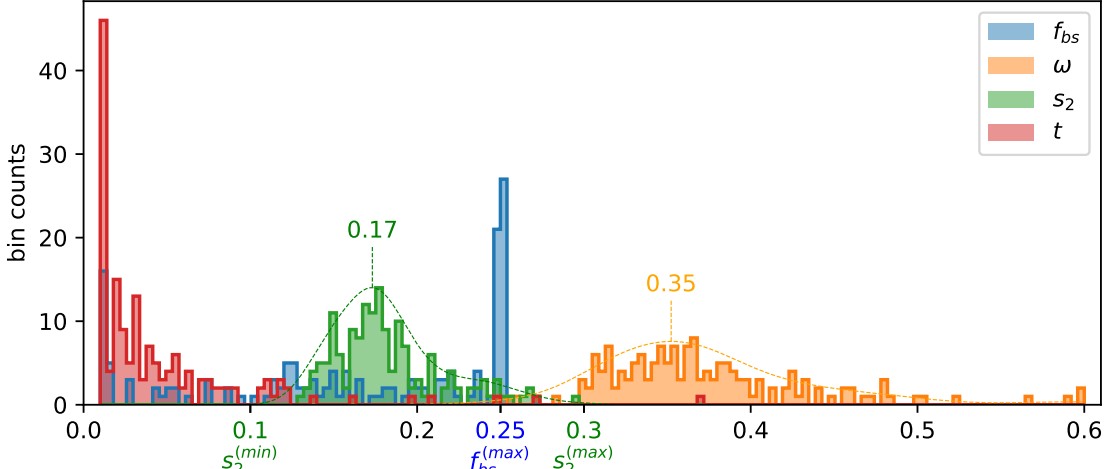

**Figure 12.** Histogram of the resulting parameter-values. (Lower and upper bounds used in the retrieval-procedure that deviate from 0. and 0.6 are indicated explicitly.)

The Pearson-correlation coefficients ($R^2$) of retrieved vs. auxiliary *SM* timeseries (2010–2012) for all sites are depicted in Figure 13. It can be seen that a good overall correlation is obtained where a slight increase in $R^2$ can be observed when auxiliary LAI timeseries are used to force the temporal variability of $\tau$ compared to a simultaneous retrieval of both $\tau$ and *SM*. Due to the fact that the model-calibration (from 2007–2009) has already been performed with the SURFEX *SM* dataset, the observed bias between retrieved and auxiliary 3-year mean *SM* values is expectedly low, with a maximum of 0.029 (mean 0.01) for *SM* retrieval with auxiliary LAI input and 0.049 (mean 0.014) for simultaneous *SM* and $\tau$ retrieval. To assess the origin of low $R^2$ values, the results in Figure 13 are color-coded with respect to the topographic complexity ($c_{topo}$) (defined as the normalized standard-deviation of elevation within an ASCAT-pixel (see Appendix A)). Most of the lowest $R^2$ values (as well as most unusually high $\omega$-values) are found in regions with high $c_{topo}$, a result that can be understood by looking at the following effects associated with $c_{topo}$:

- High $c_{topo}$ is generally observed in mountainous regions which also show high percentage of freezing-periods throughout the year. Therefore, a high percentage of the data is masked during processing, which results in a distortion (or disappearance) of the seasonality within the $\sigma_0$ timeseries.
- A $\sigma_0$ measurement of a scene with high $c_{topo}$ represents an average of reflections originating from strongly fluctuating topographies. Since $\sigma_0$ is highly dependent on the incidence-angle $\theta_0$, the obtained pair of ($\sigma_0$, $\theta_0$)-values is consequently hardly representative for the actual measurement.

In the first step, both effects lead to ambiguities in the calibration-procedure. For the hereby used model-parametrization, this results in unusually high $\omega$-values. In the second step, the performance of the *SM* retrieval is diminished due to the usage of inconsistent model-parameters alongside the inherent issues with the representativeness of the used ($\sigma_0$, $\theta_0$) pairs for sites with high $c_{topo}$.

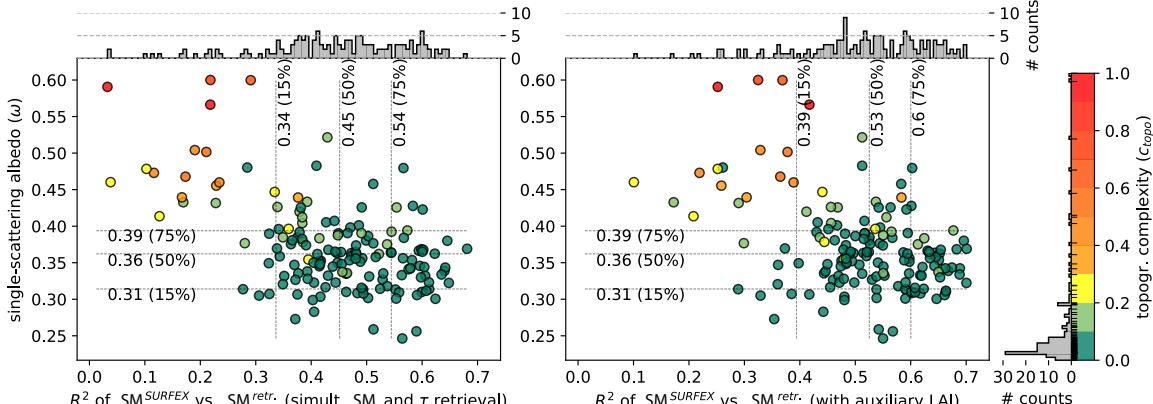

**Figure 13.** Dependency of $R^2$ for retrieved vs. SURFEX $SM$-timeseries (with and without the use of auxiliary LAI datasets) to $\omega$. To highlight the impact of topography on the obtained $\omega$- and $R^2$-values, results are color-coded with respect to $c_{topo}$. The labelled lines indicate the 15, 50 and 75 percentiles.

Considering the $\tau$ retrievals, a simple evaluation of its $R^2$ values to auxiliary LAI datasets is not meaningful, since the obtained $\tau$-timeseries generally show a high period-to-period variability and the associated seasonalities (obtained via rolling mean values) exhibit time-shifts of several days to several months with respect to the auxiliary LAI datasets. However, some insights can be drawn by looking at the behaviour of spatially averaged timeseries over all considered sites. The resulting $SM$- and $\tau$ timeseries for retrieval with auxiliary LAI- (or $SM$) inputs are shown in Figure 14a. It is found that the spatially averaged $SM$ timeseries generally show a good agreement with the averaged auxiliary $SM$ dataset. In contrast, the resulting $\tau$ timeseries exhibits a time-shift of several months with respect to the auxiliary LAI dataset, where the maximum is located around April/May. At a closer look, the obtained differences of retrieved $\tau$ and auxiliary LAI are however strongly dependent on differences between retrieved $SM$ (with aux. LAI input) and auxiliary $SM$. In fact, the retrieved $SM$ timeseries (with aux. LAI input) generally suggests lower soil-moisture compared to the auxiliary $SM$ dataset from March to June. Since changes in $SM$ have a higher impact on $\sigma_0$ than changes in $\tau$, the effect is much more pronounced in the resulting $\tau$ timeseries.

The collocation of $SM$- and $\tau$ differences suggests that the origin of the unexpectedly early peak in the $\tau$ timeseries is a systematic effect that stems from inconsistencies within the used auxiliary datasets from March to June. Since in Figure 14a, either $SM$ or $\tau$ has been provided as auxiliary dataset, any inconsistencies will lead to an erroneous subdivision of the ASCAT $\sigma_0$ dataset into soil- vegetation- and interaction contributions. The resulting agreement between modelled and measured $\sigma_0$ (Figure 11) indicates that the auxiliary $SM$ dataset is not sufficiently representative to be used as an input-dataset for $\tau$ retrieval since even minor discrepancies might have a considerable impact on the retrieved $\tau$ values. On the other hand, the auxiliary LAI dataset adequately represents the temporal dynamics of the vegetation-coverage, leading to a good agreement of the modelled $\sigma_0$ dataset as well as the resulting $SM$ timeseries. When looking at the combined $SM$- and $\tau$ retrieval-results (Figure 14b), a very similar $SM$ timeseries is found, where the lower values between March-June are still present. The associated $\tau$ timeseries now has its maximum around June-July, similar to an auxiliary LAI-dataset where exclusively grasslands have been considered (indicated by the dashed line).

The origin of the repeatedly observed lower $SM$ values in Spring however remains subject to research, where possible explanation-models are:

- The topmost soil-layer sensitive to C-band microwave observations in fact exhibits lower $SM$ values during Spring compared to the SURFEX reference dataset.
- The retrieval of low $SM$ values is actually caused by structural changes in the scattering behaviour of the vegetation-coverage (or the soil-layer) which have not been accounted for in the used model parametrization.

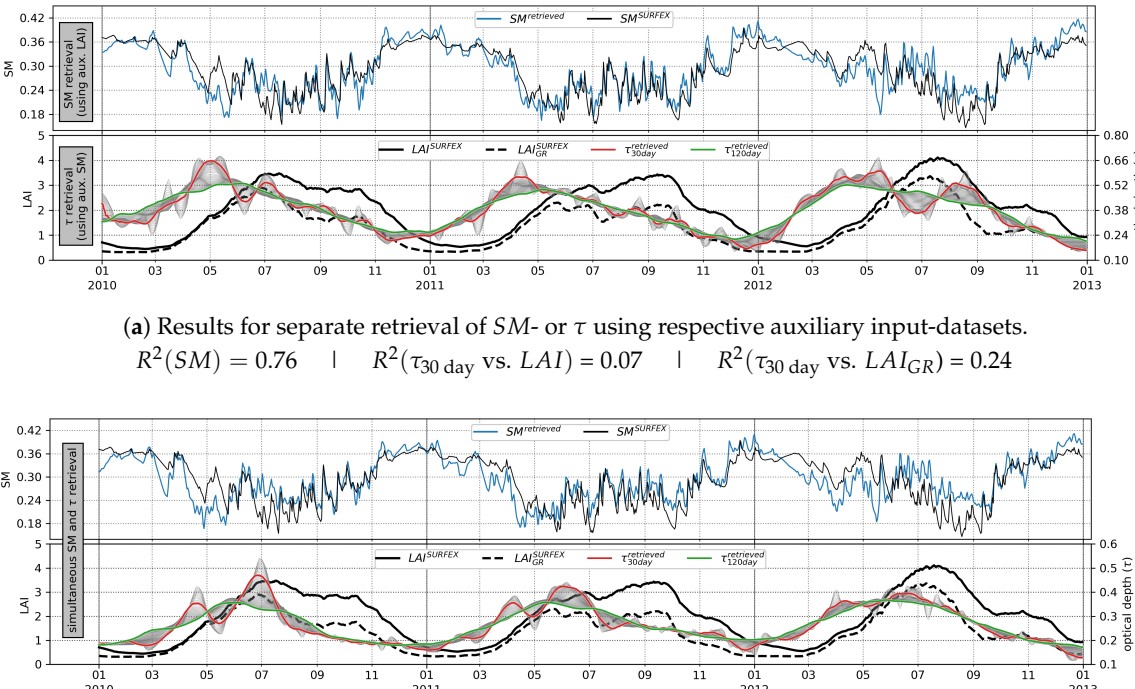

(**a**) Results for separate retrieval of *SM*- or $\tau$ using respective auxiliary input-datasets.
$R^2(SM) = 0.76$    |    $R^2(\tau_{30\,day}$ vs. $LAI) = 0.07$    |    $R^2(\tau_{30\,day}$ vs. $LAI_{GR}) = 0.24$

(**b**) Results for simultaneous retrieval of both *SM*- and $\tau$ without auxiliary datasets.
$R^2(SM) = 0.62$    |    $R^2(\tau_{30\,day}$ vs. $LAI) = 0.3$    |    $R^2(\tau_{30\,day}$ vs. $LAI_{GR}) = 0.57$

**Figure 14.** Retrieval results for *SM*- and $\tau$ in comparison to auxiliary *SM*- and LAI datasets. Each timeseries corresponds to a spatial average over all processed sites. The gray shading in the $\tau$ retrievals represents the variability of the results with respect to different choices of rolling-mean value periods from 2 to 120 days. The dashed line ($LAI_{GR}^{SURFEX}$) represents an LAI timeseries where exclusively grasslands have been considered.

To provide a closer insight in the retrieval-procedure and the associated results, Figure 15 shows details for a single selected site. The top images depict the incidence-angle behaviour of the ASCAT $\sigma_0$ dataset from 2010–2012 together with the resulting modelled $\sigma_0$ values and its separation into soil-vegetation- and interaction-contributions. Below, the very same dataset is visualized as a time-series, where the emerging seasonality of the individual contributions and their respective share in the modelled $\sigma_0$ dataset becomes visible.

The corresponding *SM* and $\tau$ datasets obtained from the retrieval-procedure are shown in the last two graphs. For this specific point, the seasonality of $\tau$ closely follows the seasonality of the auxiliary LAI dataset. The high variability of the retrieved $\tau$ estimates indicate that a simultaneous retrieval of both *SM* and $\tau$ suffers from ambiguities since in certain time-periods, the effects of a change in *SM* or $\tau$ on $\sigma_0$ cannot be satisfactorily distinguished based on the available ASCAT $\sigma_0$ measurements. However, additional constraints concerning the allowed numerical range of *SM* and $\tau$ as well as a more rigorous selection of time-periods used in the $\tau$-retrieval bears potential for improving the obtained separation of *SM* and $\tau$.

To illustrate how changes in *SM* and $\tau$ actually affect the modelled $\sigma_0$ dataset, Figure 16 shows daily retrieval details for August 2010 (of the same site as in Figure 15). The top figure-rows show the available ASCAT $\sigma_0$ measurements and the resulting modelled $\sigma_0$ graphs for 2 consecutive days (indicated by red and black colors), and the resulting *SM*- *LAI*- and $\tau$ timeseries are depicted below.

Analysis of such plots serves as a tool for deducing possible origins for ambiguity in the retrieval-process as well as for optimizing the representations of the used scattering-distributions to better match the observed incidence angle behaviour and day-to-day dynamics for a given site.

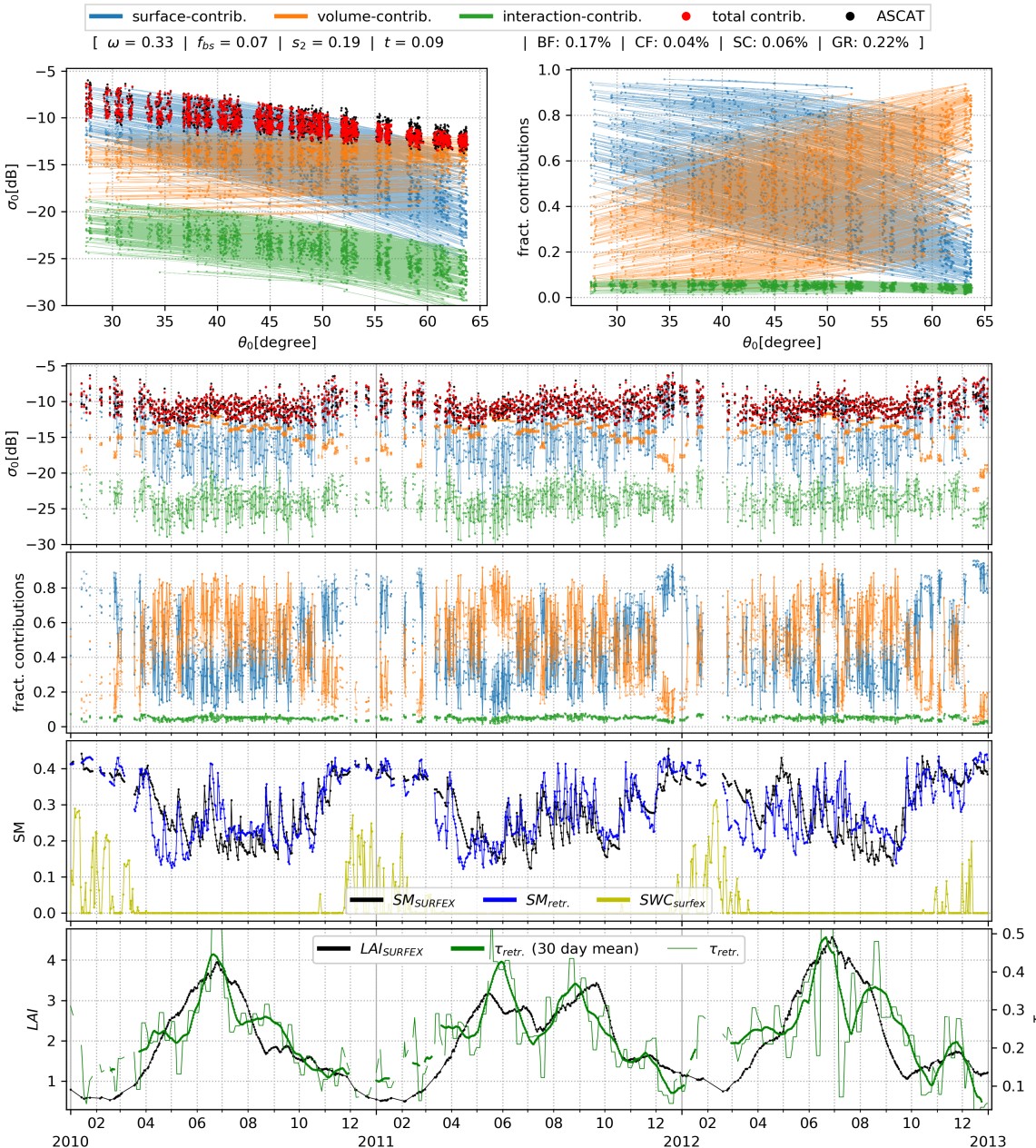

**Figure 15.** Details on the retrieval results of a single site for simultaneous retrieval of *SM* and $\tau$. The site is located at (lat/lon) = (45.83/0.7) (see Figure 10). Used fractional vegetation-coverages as well as the resulting model parameters are indicated below the legend. The first images show the incidence-angle dependency of the ASCAT $\sigma_0$ measurements as well as the modelled $\sigma_0$ dataset and its associated separation into soil-, vegetation- and interaction-contributions. Below, the very same dataset is depicted as timeseries together with the resulting *SM*- and $\tau$ timeseries as well as associated auxiliary *SM*- and *LAI* datasets. The additional SWC timeseries depicts the solid-water content dataset used for masking frozen-soil conditions.

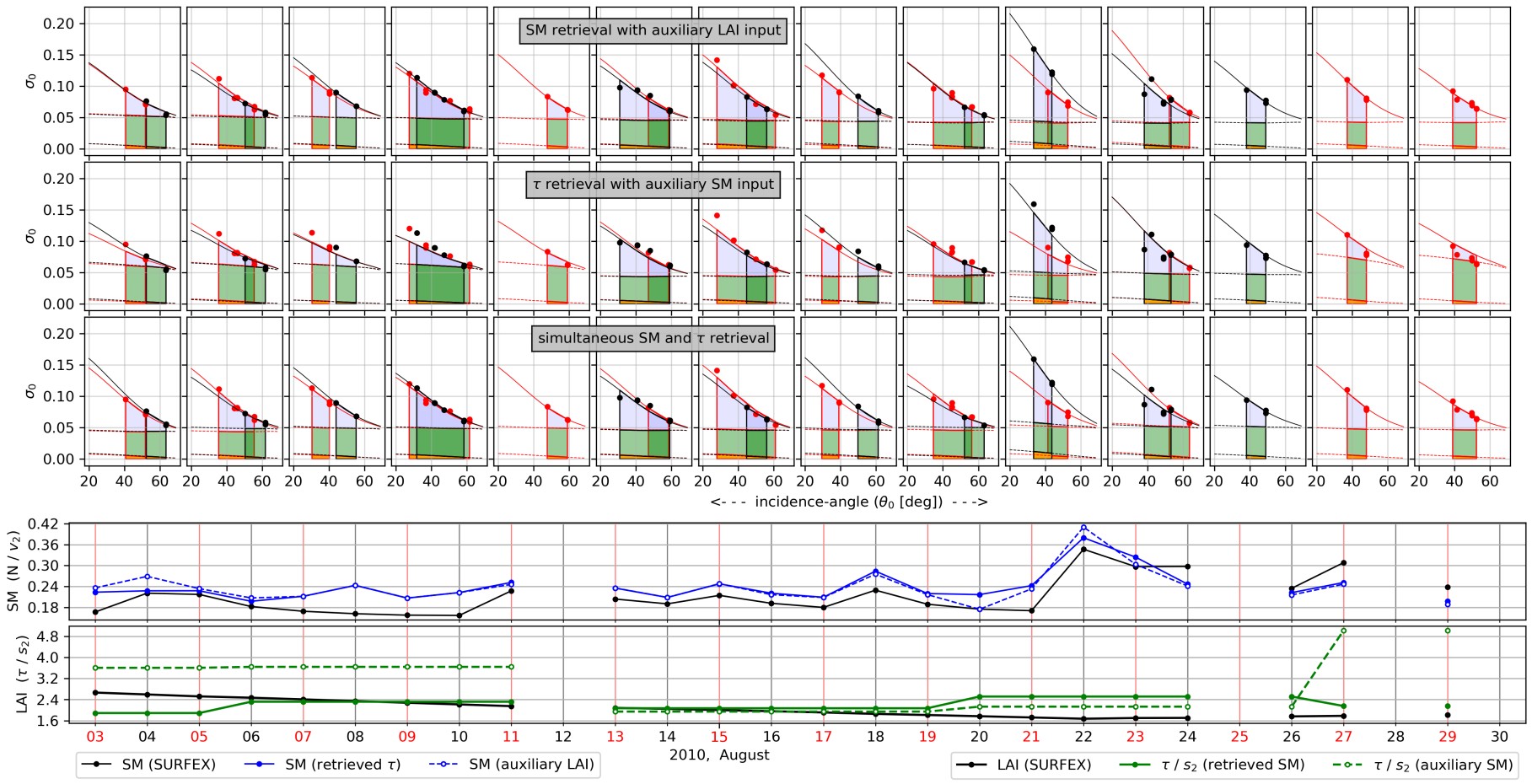

**Figure 16.** Daily retrieval details for August 2010 of the same site as in Figure 15. The first three rows show ASCAT $\sigma_0$-measurements (dots) together with the resulting separation into soil- (blue) vegetation- (green) and first-order interaction-contributions (orange). Each plot illustrates the results of 2 consecutive days, identifiable by the label-colouring of the *SM*- and *LAI*-timeseries below.

## 5. Conclusions

In this paper we introduced a radiative-transfer based modelling framework for satellite-borne (microwave) $\sigma_0$ measurements of vegetated terrain. We showed that the presented modelling approach can successfully be used for forward-simulation and soil-moisture inversion of incidence-angle dependent ASCAT $\sigma_0$ measurements over a set of 158 sites within France. The hereby used model-parametrization incorporates only two temporally varying parameters, namely soil-moisture ($SM$) and vegetation optical depth ($\tau$). Resulting $SM$- and $\tau$-timeseries are compared to auxiliary $SM$- and $LAI$ datasets to assess the overall retrieval performance as well as possible issues concerning the representativeness of used auxiliary datasets. The high flexibility of the functional representations for the scattering distributions can be used to adjust the model to various land-cover types without changing the principal modelling framework. This allows generating comparable sets of models that represent a wide range of soil- and vegetation types. Analysis of the different contributions to the total backscatter provides a deeper understanding on the mechanisms that lead to the measured backscattering signal of scatterometer measurements. While the presented application of the model has been based on an empirical selection of parametric scattering distributions, the investigation of a-priori specifications of the scattering distributions based on soil- and vegetation characteristics as well as an incorporation of its seasonal variations via temporally varying parameters might provide valuable improvements to soil-moisture and/or vegetation optical depth retrievals.

**Author Contributions:** Conceptualization, R.Q., C.A., J.-C.C. and W.W.; Data curation, R.Q. and C.A.; Formal analysis, R.Q.; Methodology, R.Q.; Software, R.Q.; Visualization, R.Q.; Writing—original draft, R.Q.; Writing—review & editing, R.Q., C.A., J.-C.C. and W.W.

**Funding:** R.Q. has been supported by a Karl Neumaier PhD scholarship.

**Conflicts of Interest:** The authors declare no conflict of interest.

## Abbreviations

The following symbols and abbreviations are used in this manuscript:

| | |
|---|---|
| ASCAT | Advamced Scatterometer |
| ISBA | Interactions between Soil, Biosphere and Atmosphere land-surface model |
| SURFEX | Surface Externalisée (in French), (a surface modelling platform developed by Météo-France) |
| BF | Broadleaf Forest |
| CF | Coniferous Forest |
| SC | Straw Cereals |
| GR | Grasslands |
| $I_{inc}, I_s$ | (incident, scattered) intensity |
| $\sigma_0$ | Normalized backscattering coefficient $\left(\sigma_0 = 4\pi \cos(\theta) \frac{I_s}{I_{inc}}\right)$ (see (1) and [54]) |
| $\sigma_0^s$ | $\sigma_0$ contribution originating from bare soil scattering |
| $\sigma_0^v$ | $\sigma_0$ contribution originating from scattering events within the vegetation-layer |
| $\sigma_0^{vs}$ | 1st order scattering contribution to $\sigma_0$ (vegetation-layer $\rightarrow$ soil-surface $\rightarrow$ detector) |
| $\sigma_0^{sv}$ | 1st order scattering contribution to $\sigma_0$ (soil-surface $\rightarrow$ vegetation-layer $\rightarrow$ detector) |
| $\sigma_0^{int}$ | 1st order interaction-contribution ($\sigma_0^{int} = \sigma_0^{vs} + \sigma_0^{sv}$) |
| $\Theta$ ($\Theta_a$) | (generalized) scattering angle (see (5) and (6)) |
| $\theta, (\theta_0, \theta_s)$ | polar zenith-angle (of incident, scattered radiation) in a spherical coordinate system |
| $\phi(\phi_0, \phi_s)$ | azimuth-angle (of incident, scattered radiation) in a spherical coordinate system |
| $\tau$ | Optical depth (see [11]) |
| $\omega$ | Single scattering albedo (see [11]) |
| $\gamma$ | 2-way attenuation factor $\left(\gamma = e^{-\frac{\tau}{\cos(\theta)}}\right)$ |
| $\hat{p}$ | Volume-scattering phase-function (see Section 2.1.2) |
| $BRDF$ | Bidirectional reflectance distribution function (see Section 2.1.3) |
| $R(\theta, \phi)$ | Hemispherical Reflectance (see (9)) |
| $t$ | Asymmetry-factor of the used $BRDF$-representation (see (9)) |
| $LAI$ | Leaf area index |
| $v_2$ | Scaling factor between $\tau$ and $LAI$ (see (12)) |
| $SM$ | Volumetric soil-moisture content |

$N$      Nadir hemispherical reflectance of *BRDF* defined in Section 2.1.3
$s_2$      Scaling factor between $N$ and $SM$ (see (13))
$f_{bs}$      Effective bare-soil fraction (see Sections 2.1 and 3.5)
$HG$      Henyey-Greenstein function (see (3))
$R^2$      (squared) Pearson correlation coefficient
$c_{topo}$      Topographic complexity (see Appendix A)

## Appendix A. Topographic Complexity ($c_{topo}$)

The topographic complexity $c_{topo}$ is introduced as a measure of topographic variability within the footprint. $c_{topo}$ is calculated by evaluating the 2d-hamming-window weighted standard-deviations of the elevation-data provided by the GTOPO30 DEM (courtesy of the U.S. Geological Survey (https://usgs.gov)) within an approx. 24 km$^2$ grid (to obtain values comparable to ASCAT $\sigma_0$ footprints, see [55]). The resulting values of $c_{topo}$ for the considered sites are then normalized to range between 0 and 1.

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
