# Peer review of "A Generic First-Order Radiative Transfer Modelling Approach for the Inversion of Soil and Vegetation Parameters from Scatterometer Observations"

_remotesensing, doi:10.3390/rs11030285_

Round 1
Reviewer 1 Report
2nd Review
JournalRemote Sensing (ISSN 2072-4292)
Manuscript ID remotesensing-407584, remotesensing-438838
Type Article
Number of Pages 24
Title A generic first-order Radiative Transfer modelling approach for the inversion of soil- and vegetation parameters from scatterometer observations.
The Author did answer all questions sufficiently. I dont have any further comment. As mentined in the first review the paper is well written and comprehensive. A comparison to a theoretical model would be nice, but I think it would extend the already long manuscript to infinity. This could be part of a 2nd publication. In addition as the author answered in length that this manuscript is an introduction of a general modelling-approach and a parametrization that the behaviour in the backscattering direction follows the behaviour of ASCAT measurements. One could question if the title is a bit missleading and if the described method is an inversion, but in the end its a mixure of empirical modelling, paramerization and inversion. I am still with the manuscript and would propose to publish it.
Reviewer 2 Report
The approach is novel and interesting to the readers, and that is why I believed a clear description of its limitation was needed for its future development/applications. The current version has improved over the previous submission in terms of clarity and accuracy. I encourage the authors extend the work to the operational level in next studies.
This manuscript is a resubmission of an earlier submission. The following is a list of the peer review reports and author responses from that submission.
Round 1
Reviewer 1 Report
The study tries to solve the forward modelling and inversion problems from a different approach than the traditional microwave remote sensing means. It is always encouraging for such studies, but this work is not satisfying in a few aspects as follows:
(a) I am confused if the work emphasized modelling or inversion. If focused on modelling, the authors need to demonstrate the performance of the new empirical model by comparing with theoretical models. I understand this work was designed for complex scenario, but it should be applied to/validated against relatively simple situations first. For example, can you compare with theoretical surface scattering model before plotting Fig. 5?
(b) Line 162-166: “In contrast to … that the amount of scattered radiation in specular direction originating from a (perfectly smooth) surface is increasing with increasing incidence-angle”. For vertical polarization, isn’t the microwave reflectivity decreases with incidence angles until reaching the Brewster angle for the air-soil surface?
(c) The authors used ISBA outputs for training and validation, which cannot be deemed as effective evaluations of the effectiveness of the approach in inversion problems due to uncertainties associated with ISBA; and cannot be applied elsewhere than France. If the study focused on inversion, then in-situ measurements are a must.
(d) Line 44: for crops, what do you think about the impacts of changing structures and biomass on vegetation scattering phase matrix or phase function? Will there be high uncertainties by assuming most parameters in the phase functions “constant over the considered time-period”?
(e) I do not think Eq. (12) works well for tress since it does not represent trunks well.
(f) “The fact that no significant increase in correlation can be obtained in the τ-retrieval indicates that the modelled σ0 dataset is less sensitive to changes in τ compared to changes in SM”. For forests, can this statement be valid considering C-band penetration ability?
(g) Line 439: Any snow in March?
(h) I suggest to shorten the paper and remove some figures. For example, the upper three rows of Fig. 15 do not provide much information to the readers.
The comments might be critical, but overall this is an interesting research and a good exploration on microwave modelling and inversion. I would like to read an improved version.
Author Response
Response to the comments from reviewer 1 for the manuscript (ID: remotesensing-407584) “A generic first-order Radiative Transfer modelling approach for the inversion of soil- and vegetation parameters from scatterometer observations.” by Raphael Quast, Clément Albergel, Jean-Christophe Calvet and Wolfgang Wagner.
First of all I want to thank the reviewer for the valuable comments and suggestions. In the following a list addressing all comments of the review is given:
(a) I am confused if the work emphasized modelling or inversion. If focused on modelling, the authors need to demonstrate the performance of the new empirical model by comparing with theoretical models. I understand this work was designed for complex scenario, but it should be applied to/validated against relatively simple situations first. For example, can you compare with theoretical surface scattering model before plotting Fig. 5?
With the presented work we tried to combine both modelling and inversion in order to assess aspects of coarse resolution soil-moisture retrieval that are hardly accessible from a purely theoretical point of view:
The hereby used phase-functions represent areas of tens of kilometres with varying vegetation-coverage (plant-type, density, vegetation water content...) and soil-properties (roughness, texture, moisture ... ) An a-priori parametrization based on theoretical considerations is therefore hardly possible.
The parametrization of the soil-surface was chosen as an example that provided good results within the presented study. It is merely to be seen as a parametric fit-function rather than a theoretical soil-scattering model.
We avoided a comparison to theoretical soil-scattering models due to several issues in the parametrization of such models for spatial resolutions in the order of kilometres such as:
Using theoretical models like the I2EM requires the specification of the statistical roughness-distributions. This choice has a huge impact on the results, but there is no clear suggestion on what would be an adequate choice for natural surfaces at C-band
Estimation of RMS-height and correlation-length is generally performed based on the visual appearance of a surface (i.e. in the optical domain)... a direct application of the obtained values to microwave observations of natural surfaces seems rather ambiguous.
Topographic variations within the observed scene might also contribute a considerable effect when dealing with spatial resolutions in the order of kilometres
I agree that an incorporation of theoretical findings into the parametrization of the phase-functions would add valuable insights, but concerning the aforementioned issues, such investigations unfortunately lie beyond the scope of this paper.
However, some quick calculations (using the implementation of the I2EM provided by https://github.com/ibaris/pyrism) supports a behaviour similar to the proposed one (considering either Gaussian or exponential roughness distributions, RMS of 25px and corr. length of 5-10 cm) i.e.:
the BRDF is represented as a peak in specular direction
The width of the peak however is generally more directional than the BRDF used in the model. This discrepancy can be explained by the fact that the model-BRDF represents very large, inhomogeneous areas while the I2EM simulations are results for a specific (homogeneous) type of surface.
the magnitude of the BRDF in specular direction is increasing with increasing incidence angle
for VV polarization this increase is only until a certain incidence-angle that is generally higher (>60 degree) than the maximum incidence-angle of the ASCAT instrument. Since any direction away from the monostatic direction contributes only to the estimation of the interaction-contribution, errors in the representativeness of the BRDF at those angles will have minor effects on the simulated backscattered radiation.
(b) Line 162-166: “In contrast to … that the amount of scattered radiation in specular direction originating from a (perfectly smooth) surface is increasing with increasing incidence-angle”. For vertical polarization, isn’t the microwave reflectivity decreases with incidence angles until reaching the Brewster angle for the air-soil surface?
You are of course right. Considering the reflection of a flat surface this statement is only valid for horizontally polarized radiation. The main reason for stating this was to emphasize that the incidence-angle dependency of the BRDF of a surface requires a different treatment than the incidence-angle dependency of a volume scattering phase function (which was assumed to be isotropic).
The sentence has been changed to:
"...the behaviour of Fresnel's reflection coefficients indicate that the amount of scattered radiation in specular direction originating from a (perfectly smooth) surface has a complex (polarization dependent) incidence-angle behaviour."
(c) The authors used ISBA outputs for training and validation, which cannot be deemed as effective evaluations of the effectiveness of the approach in inversion problems due to uncertainties associated with ISBA; and cannot be applied elsewhere than France. If the study focused on inversion, then in-situ measurements are a must.
The presented study introduced the modelling approach and presented a rather specific application in order to show that the approach is in fact capable of representing incidence-angle dependent ASCAT sigma_0 measurements based on soil- and vegetation dynamics. It is not yet intended as an operational method.
Even though the training involved ISBA outputs, the result of the calibration-procedure are temporally constant parameters obtained from a time-period that does not overlap with the retrieval period.
The parameters characterize the spatial variability of soil- and vegetation properties (that are hardly accessible by other means) by adjusting the impact of a change in soil-moisture (and/or vegetation optical depth) on the backscattered radiation.
Once the parameters are obtained, the soil-moisture dynamics are solely obtained from the dynamics of the ASCAT signal. (except for one of the inversion experiments where ISBA LAI estimates are used as an auxiliary dataset). The generally high correlations to the ISBA simulations within the inversion period suggest that there is a close connection between the ISBA output and the signal measured by the ASCAT instrument.
Furthermore, ISBA is a generic large-scale land surface model that can be run offline anywhere providing that atmospheric forcing is available. Several studies have focused on the use of ISBA forced e.g. by ERA-Interim and ERA5 atmospheric reanalysis from ECMWF at e.g. continental scale (Leroux et al., 2018 over Europe, Albergel et al., 2018 a&b over North America). See references below.
Leroux, D.J.; Calvet, J.-C.; Munier, S.; Albergel, C. Using Satellite-Derived Vegetation Products to Evaluate LDAS-Monde over the Euro-Mediterranean Area. Remote Sens. 2018, 10, 1199.
Albergel, C., Dutra, E., Munier, S., Calvet, J.-C., Munoz-Sabater, J., de Rosnay, P., and Balsamo, G.: ERA-5 and ERA-Interim driven ISBA land surface model simulations: which one performs better?, Hydrol. Earth Syst. Sci., 22, 3515-3532, https://doi.org/10.5194/hess-22-3515-2018, 2018.
Albergel, C.; Munier, S.; Bocher, A.; Bonan, B.; Zheng, Y.; Draper, C.; Leroux, D.J.; Calvet, J.-C. LDAS-Monde Sequential Assimilation of Satellite Derived Observations Applied to the Contiguous US: An ERA-5 Driven Reanalysis of the Land Surface Variables. Remote Sens. 2018, 10, 1627.
(d) Line 44: for crops, what do you think about the impacts of changing structures and biomass on vegetation scattering phase matrix or phase function? Will there be high uncertainties by assuming most parameters in the phase functions “constant over the considered time-period”?
In general one must always consider that the area under investigation is approximately a circle with a radius of 12.5 km. Therefore, when considering the phase-function as an average over the whole observed scene, the assumption of constant parameters seemed reasonable.
Aside of the question of reasonability however, there is also the question of accessibility of the information that would be required in order to parametrize the phase-functions in a dynamic way. While the overall framework presented in the paper could be extended to include also temporal changes in the phase-functions, the available (daily) measurements from the ASCAT instrument do not allow an unambiguous retrieval of additional temporally varying parameters.
We therefore tried to simulate the temporal dynamics of the vegetation coverage solely via the temporal dynamics of the LAI while structural changes (reflected in the single-scattering albedo and the additional parameters) are assumed to vary only spatially.
I agree that from a theoretical perspective the incorporation of temporally changing structures and biomass seems important, however from a practical point of view there is simply no data available that would support such a parametrization on a spatial scale of tens of kilometres.
(e) I do not think Eq. (12) works well for tress since it does not represent trunks well.
Again I fully agree that Eq.12 not even closely represents the complexity that is encountered within the observed scene. The question if there exist better parametrizations and how they might relate to the actual vegetation coverage is very interesting, but lies beyond the scope of this paper. The intention here was to use rather simple relations in order not to increase the amount of unknown parameters.
Concerning the presented study, forests contribute only a fraction of 10-20% to the average vegetation-coverage of the considered sites. Thus, there is a high possibility that the contributions of trunks to the backscattered radiation are negligible.
(f) “The fact that no significant increase in correlation can be obtained in the τ-retrieval indicates that the modelled σ0 dataset is less sensitive to changes in τ compared to changes in SM”. For forests, can this statement be valid considering C-band penetration ability?
As stated above, forests contribute only a fraction of 10-20% to the average vegetation-coverage encountered within the considered sites. Therefore we expect a significant portion of the illuminated area to be covered with no or moderately dense vegetation for which the C-band penetration ability will be sufficient.
(g) Line 439: Any snow in March?
There are several possibilities that might lead the ISBA soil-moisture values to be higher than the retrieved ones. Snow and soil-freezing are of course always affecting microwave soil-moisture retrievals but due to the following points I would not directly connect this result to snowfall:
Fig. 14 shows an average over all processed sites (located throughout France)
days with non-zero solid-water content in the ISBA simulations have already been masked
the lower soil-moisture retrievals are present till March-June
What we wanted to stress in here is that the vegetation-optical-depth retrievals (with auxiliary ISBA-soil-moisture input) are very sensitive to the auxiliary dataset since a change in soil-moisture has a much greater impact on the backscattered radiation than a change in vegetation-optical-depth.
(h) I suggest to shorten the paper and remove some figures. For example, the upper three rows of Fig. 15 do not provide much information to the readers.
In my opinion the first three rows of Fig. 15 provide a rarely seen insight into the raw-data that underlies a soil-moisture retrieval from active microwave observations which is possibly very useful to readers that are not familiar with radar datasets.
The figures furthermore depict the obtained decomposition of the signal into contributions originating from soil- and vegetation, both with respect to angular and temporal variations, which is otherwise only evident within the formulas.
Since I personally think that the figures help to make the approach accessible to a broader audience, I would prefer keeping it as is.

Reviewer 2 Report
Review MDPI Remote Sensing
A generic first-order Radiative Transfer modelling approach for the inversion of soil- and vegetation parameters from scatterometer observations.
Raphael Quast * , Clément Albergel , Jean-Christophe Calvet , Wolfgang Wagner
The Authors present an empirical radiative transfer model to derive coarse resolution soil moisture (SM) and vegetation optical depth from ASCAT microwave backscatter measurements. The model was tested and applied to 158 selected test sides within France together with different auxiliary datasets of Leaf area index (LAI) and SM.
General comment:
In general the manuscript is well defined, easy to read and with a good structure. Only minor changes should be considered.
Abstract:
7: I would name the vegetation optical depth Tv instead of T
Figur 1: An arrow of incident angle is missing. Where are the angles of Theta0 ?
Figure 2: Where are the angles of Thetas Theta0 , Phis, Phi0 ?
273: VOD?
276-297: I dont see that much differences in SSA estimates? [36]: 0.4-0.6, [37]: 0.3-0.8, [38]: 0.4-0.6
296: WCM?
297: Please describe Gamma.
298: The term p(Theta0 → Theta0) is not clear to me in this context.
347: What is the parameter fbs and where did this come from? I didnt see it in the previous sections.
372-373: Why most? I would recommend to check if all necessary routines are available to understand and reproduce the results of the manuscript.
Figure 12: Legend hard to read.
423-451: Hard to understand without a comprehensible Figure 14.
Figure 14: Dont see any graph.
Author Response
Response to the comments from reviewer 2 for the manuscript (ID: remotesensing-407584) “A generic first-order Radiative Transfer modelling approach for the inversion of soil- and vegetation parameters from scatterometer observations.” by Raphael Quast, Clément Albergel, Jean-Christophe Calvet and Wolfgang Wagner.
First of all I want to thank the reviewer for the valuable comments and suggestions. In the following a list addressing all comments of the review is given:
7: I would name the vegetation optical depth Tv instead of T
The tau-symbol is used exclusively to denote the vegetation optical depth within the paper. Since there is no source of confusion we would prefer keeping the notation as is.
Figure 1: An arrow of incident angle is missing. Where are the angles of Theta0 ?
Figure 2: Where are the angles of Thetas Theta0 , Phis, Phi0 ?
Including both polar- and zenith angles in the figures would require 3D graphics.
The meaning of theta_0, theta_s and phi_0, phi_s is explained in the text as the incident- and emerging zenith- and azimuth-angles. Since those terms are clearly defined and commonly used throughout literature, an explicit depiction of both azimuth- and zenith angles in Figure 1 and 2 has been omitted.
However, an arrow corresponding to the direction of incident radiation has been added to Fig. 1 and indicators for the zenith-angles have been added to Fig. 1 and 2.
273: VOD?
Since this abbreviation is used only once, it has been changed to "vegetation optical depth".
276-297: I don’t see that much differences in SSA estimates? [36]: 0.4-0.6, [37]: 0.3-0.8, [38]: 0.4-0.6
We address here the difference between theoretical SSA estimates (generally found between 0.3 and 0.9) and the values commonly used within soil-moisture retrieval algorithms (generally termed "effective SSA" and found between 0.05 and 0.15 [35]).
I've tried to clarify this by changing the paragraph to:
... Within the literature, many different estimates on the numerical value of the single scattering albedo "omega" can be found. Retrieved values (almost entirely based on passive observations) are generally found in the range "0.05 < omega < 0.15" [35]. Theoretical models that estimate "omega" from assumptions on geometrical and dielectrical properties of the vegetation-constituents usually obtain much higher values. For example ...
296: WCM?
Changed to "water cloud model"
297: Please describe Gamma.
Definition of "Gamma" added to Eq.44
298: The term p(Theta0 → Theta0) is not clear to me in this context.
Changed to the term to "p_back" and added the following explanation of the meaning directly below the equation:
... where p_back denotes the value of p in the backscattering direction
(i.e. theta_s = theta_0 and phi_s = pi + phi_0 )
347: What is the parameter fbs and where did this come from? I didn’t see it in the previous sections.
"f_bs" appears in (Eq.1) as well as in (Fig.1) as the fraction of vegetation-covered soil and is also mentioned in the text introducing the overall modelling approach (section 2.1).
Therefore no further explanations have been added.
372-373: Why most? I would recommend to check if all necessary routines are available to understand and reproduce the results of the manuscript.
The term "most" has been used since the python-module is still under development and lacks on a (user-friendly) documentation for some parts.
With sufficient knowledge on how the module works, all results can be reproduced with the provided code.
I've changed the sentence to:
The utilized routines are available as part of the...
Figure 12: Legend hard to read.
The legend has been changed to:
Histogram of the resulting parameter-values.
(Lower and upper bounds used in the retrieval-procedure
that deviate from 0. and 0.6 are indicated explicitly)
423-451: Hard to understand without a comprehensible Figure 14.
Figure 14: Don’t see any graph.
Unfortunately I do not fully know how to respond to this issue. Is the Figure not displayed correctly in your PDF-reader? (I can not reproduce this on my computer using Adobe Acrobat Reader or SumatraPDF to view the document)
To enhance compatibility, the pdf-figure has been changed to a high-resolution png.

Round 2
Reviewer 1 Report
Here are my comments for the revised version:
(a) Line 172-173 and the t=0.6 curve of the right figure of Fig. 5.
Again, the angular pattern of scattering depends on polarization and surface roughness. It is not accurate to state “increasing with increasing incidence-angle”. What is the polarization used for Figure 5? The simulations need to reflect ASCAT configurations including its VV polarization.
(b) As mentioned in my previous comments, I believe the lack of rigid comparisons with theoretical model in the modeling part and ground measurements in the validation weaken the work significantly. Without these rigid comparisons, which were commonly practiced in other satellite studies (e.g. SMOS algorithm development and validation work), it is hard to make independent evaluations of the accuracy and effectiveness of the new approach.
(c) A 10% to 20% forest coverage means a much higher percentage of contribution from trees to radar observations. This is not "negligible" and needs to be discussed.
I like this novel approach, but suggest the authors provide revisions and a thorough discussion to address the above issues.
Author Response
(a) Line 172-173 and the t=0.6 curve of the right figure of Fig. 5.
Again, the angular pattern of scattering depends on polarization and surface roughness. It is not accurate to state “increasing with increasing incidence-angle”. What is the polarization used for Figure 5? The simulations need to reflect ASCAT configurations including its VV polarization.
The sentence containing the term "increasing with increasing incidence-angle" has already been removed in the previous version, and had been replaced with "... has a complex (polarization-dependent) incidence angle behaviour"
There is in fact no actual choice of polarization assigned to Figure 5, since it just illustrates the impact of a change in the t- and a- parameter on the incidence-angle behaviour of the parametric BRDF-function introduced in Eq. 10. However, since this function is finally used as parametric representation of the BRDF when being applied to ASCAT measurements, we expect certain combinations of a- and t- parameter values to be representative for VV-polarization.
The actual parameters are finally derived within the "forward-simulation" experiment (where the "a"-parameter has been set manually to 0.6 to avoid ambiguity in the optimization procedure). Looking at the resulting parameters, we see a very strong trend to "t"-values < 0.1. A comparison with Figure 5 shows that for t ~ 0.1 the BRDF shows a decrease of the scattered intensity in specular direction with respect to the incidence-angle, which is what one would expect also from the VV-polarized Fresnel-coefficient. (however, no Brewster-angle effects are simulated with the parametric function, so there is a monotonous decrease of the scattered intensity with respect to the incidence-angle)
To clarify that the introduced BRDF parametrization is not a theoretical soil-scattering model, but rather a parametrization that has been found to reflect the behaviour of coarse resolution scatterometer measurements reasonably well, the following paragraph has been inserted in “Section 2.1.3: Parametrization of the Bidirectional Reflectance Distribution Function”:
A comprehensive incorporation of the aforementioned properties in a functional representation of the $BRDF$ requires extensive theoretical calculations [9,10].
However, a parametric description of the $BRDF$ that can be used to represent the bare-soil contribution of coarse-resolution monostatic scatterometer measurements
within reasonable accuracy can be obtained by adjusting the previously defined generalization of the Henyey Greenstein function (6) as follows:
…
Additional complexity in the representation of the $BRDF$ (for example the incorporation of an isotropic contribution or peaks in the backscattering direction) can directly be introduced by using linear-combinations similar to (7).
Additionally the term “… of the BRDF representation introduced in (10) ” has been added to the captions of figure 4 and 5
(b) As mentioned in my previous comments, I believe the lack of rigid comparisons with theoretical model in the modeling part and ground measurements in the validation weaken the work significantly. Without these rigid comparisons, which were commonly practiced in other satellite studies (e.g. SMOS algorithm development and validation work), it is hard to make independent evaluations of the accuracy and effectiveness of the new approach.
The primary aim of this paper is the introduction of the general modelling-approach and a presentation of the possibilities of the approach with respect to a specific application.
The used parametrizations have been selected such that their behaviour in the backscattering direction follows the behaviour of ASCAT measurements.
It is stated explicitly at the end of Section 2.2.1 that the shape of the used functions in directions other than the backscattering direction is highly ambiguous but only impacts the estimation of first-order interaction effects (which are found to contribute <10% to the modelled signal).
We agree that rigid comparisons to theoretical models might serve as a valuable tool to better confine the parametric representation of the BRDF that is used to represent possible soil-types encountered within the areas of interest.
However, such studies are unfortunately beyond the scope of this paper since theoretical modelling of the scattering-behaviour of an area of 12.5 x 12.5 km with possibly inhomogeneous soil-type, roughness and topography would require extensive additional research.
In order to clarify that the presented parametrizations serve as approximation-functions that must be selected based on the specific measurement characteristics, the following passage has been added to the beginning of "Section 2.2.1: Parametrization of Scattering Distribution "
"In order to approximate the scattering-behaviour of soil- and vegetation, an empirical description of $\hat{p}$ and the $BRDF$ based on parametric functions is introduced. While the general framework can be adjusted to mimic a wide range of scattering-characteristics, the final choice for a specific application must be selected with respect to the characteristics of considered measurements (i.e. frequency, polarization, spatial resolution, ...). ...."
Additionally the beginning of “Section 3: Choice of parametrization” has been changed to the following passage to clarify the specific nature of the introduced parametrization:
“In the presented experiments (Section 4), both the choice of the functional representations of $\hat p$ and the $BRDF$ as well as the estimation of the associated parameters are based solely on measurements by the ASCAT instrument and simulations of SM and LAI provided by the SURFEX modelling platform. The hereby introduced parametrization is therefore specific to this instrument (ASCAT, see Section 2.2.1) and auxiliary-data model (ISBA forced by SAFRAN, see Section 2.2.2). The overall approach however is highly flexible and can be adjusted to various experiment-configurations. The following section illustrates the considerations that led to the selected model parametrization.“
Concerning an incorporation of ground-measurements in the validation, we similarly agree that such investigations would allow an improved assessment of the potential of the retrieval approach to estimate actual soil-moisture conditions.
However, in contrast to the SMOS algorithm, the presented model-parametrization is not intended to be used on an operational level, but rather serves as a “proof of concept” in the way that we showed that there exists a parametrization that can follow the behaviour of the ASCAT signal reasonably well while the resulting soil-moisture retrievals show a meaningful behaviour when being compared to ISBA simulations.
In order to highlight that ISBA-simulations have in-fact been validated with respect to in-situ measurements, the following paragraph has been added to Section 2.2.2: Auxiliary SM and LAI datasets:
"ISBA simulations driven by SAFRAN were compared to in situ observations of SSM in south- western France by Albergel et al. (2010) and Draper et al. (2011), using 12 stations of the SMOSMANIA (Calvet et al. 2016) observation network. The obtained Spearman temporal correlation coefficient value ranges from 0.6 to 0.8 and the median value is 0.7."
Furthermore in Section 4: Results and Discussion, we added the following sentence to state more clearly that the presented validation only indicates a close relation between ISBA-simulations and ASCAT backscattering timeseries:
"The successful calibration of the model shows that the two completely independent data-sources (i.e. SURFEX simulations and ASCAT $\sigma_0$ measurements) are consistent."
Extending the validation with a comparison to in-situ measurements unfortunately lies beyond the scope of this work due to several issues concerning ground-measurements that need to be considered and would require a significant extension of the paper such as:
• limited spatial and temporal coverage of in-situ measurements
• questions concerning the representativeness of in-situ measurements due to
o measurement-depth (extrapolation-methods to transform the derived top-layer soil-moisture signal to sensor-depth might be required)
o spatial extent (there might be considerable problems concerning the representativeness of a single in-situ station to the retrieved signal obtained from an area of approx. 12.5x12.5 km)
o sensor-placement (is the sensor located inside a forest, field, etc. ?)
o ….
Future studies using the proposed framework will certainly incorporate a validation of obtained results to in-situ measurements. The initial study-design and site-selection of such investigations however already needs to reflect the above issues in order to allow a meaningful comparison to in-situ measurements.
added References:
- Albergel, C., J.-C. Calvet, P. de Rosnay,
G. Balsamo, W. Wagner, S. Hasenauer, V. Naemi, E. Martin, E. Bazile, F.
Bouyssel, J.-F. Mahfouf, “Cross-evaluation of modelled and remotely
sensed surface soil moisture with in situ data in southwestern France”,
Hydrol. Earth Syst. Sci., 14, 2177–2191, 2010
- Calvet, J.-C., Fritz,
N., Berne, C., Piguet, B., Maurel, W., and Meurey, C.: Deriving
pedotransfer functions for soil quartz fraction in southern France from
reverse modeling, SOIL, 2, 615-629,
https://doi.org/10.5194/soil-2-615-2016, 2016
- Draper, C., Mahfouf,
J.-F., Calvet, J.-C., Martin, E., “Assimilation of ASCAT near-surface
soil moisture into the SIM hydrological model over France”, Hydrol.
Earth Syst. Sci., 15, 3829–3841, doi:10.5194/hess-15-3829-2011, 2011.
(c) A 10% to 20% forest coverage means a much higher percentage of contribution from trees to radar observations. This is not "negligible" and needs to be discussed.
I did not mean to say that the overall contributions of forests in the backscattering signal is negligible. That this is not the case can directly be seen in Figure 15. which shows that the fractional vegetation-contribution to the modelled backscatter ranges between 20% and 80% for the selected site (with a forest-coverage of approx. 19%.)
However, since the encountered forests within the areas of interest can primarily be classified as "dense-forests", we expect that the radiation (C-band) does not fully penetrate those forest-covered fractions of the pixel, and consequently contributions of trunks in the measured signal are expected to be low.